# ACT as Human: Multimodal Large Language Model Data Annotation with Critical Thinking

**Lequan Lin** [1] *   **Dai Shi** [1,2]   **Andi Han** [1,3]   **Feng Chen** [4]   **Qiuzheng Chen** [5] †

**Jiawen Li** [5]   **Zhaoyang Li** [5]   **Jiyuan Zhang** [5]   **Zhenbang Sun** [5]   **Junbin Gao** [1]

[1] University of Sydney, Australia
[2] University of Cambridge, United Kingdom
[3] Riken AIP, Japan
[4] University of Adelaide, Australia
[5] ByteDance, Australia

## Abstract

Supervised learning relies on high-quality labeled data, but obtaining such data through human annotation is both expensive and time-consuming. Recent work explores using large language models (LLMs) for annotation, but LLM-generated labels still fall short of human-level quality. To address this problem, we propose the **Annotation with Critical Thinking** (ACT) data pipeline, where LLMs serve not only as annotators but also as judges to critically identify potential errors. Human effort is then directed towards reviewing only the most "suspicious" cases, significantly improving the human annotation efficiency. Our major contributions are as follows: (1) ACT is applicable to a wide range of domains, including natural language processing (NLP), computer vision (CV), and multimodal understanding, by leveraging multimodal-LLMs (MLLMs). (2) Through empirical studies, we derive 7 insights on how to enhance annotation quality while efficiently reducing the human cost, and then translate these findings into user-friendly guidelines. (3) We theoretically analyze how to modify the loss function so that models trained on ACT data achieve similar performance to those trained on fully human-annotated data. Our experiments show that the performance gap can be reduced to less than 2% on most benchmark datasets while saving up to 90% of human costs.

## 1 Introduction

High-quality labeled data is essential for the success of supervised learning, but human annotation remains costly and difficult to scale [1, 2, 3, 4, 5]. While large language models (LLMs) have recently emerged as an alternative for data annotation [6, 7, 8, 9, 10], the labels they generate often lack the accuracy required for training reliable models [11, 12, 13]. The trade-off between annotation cost and label quality leads to our research question: **How can we incorporate LLMs into the data pipeline to efficiently reduce human cost without compromising downstream training performance?** To address this challenge, we first propose the Annotation with Critical Thinking (ACT) data pipeline. In this approach, an LLM handles the majority of annotation workloads, while a limited human budget is strategically allocated to reviewing samples flagged as potentially erroneous by another LLM-based error detector (the criticizer). Then, we provide a theoretical analysis of how to ensure that models trained on ACT data are comparable to those trained on fully human-annotated data. An illustration of the ACT data pipeline and downstream training is provided in Figure 1.

While the majority of LLM-based data pipelines solely focus on natural language processing (NLP) [12, 14, 15, 16, 17], our method further extends to visual scenarios by leveraging multimodal-LLMs

---

*The work is done during Lequan Lin's internship at ByteDance. ✉ lequan.lin@sydney.edu.au.
†Project lead.

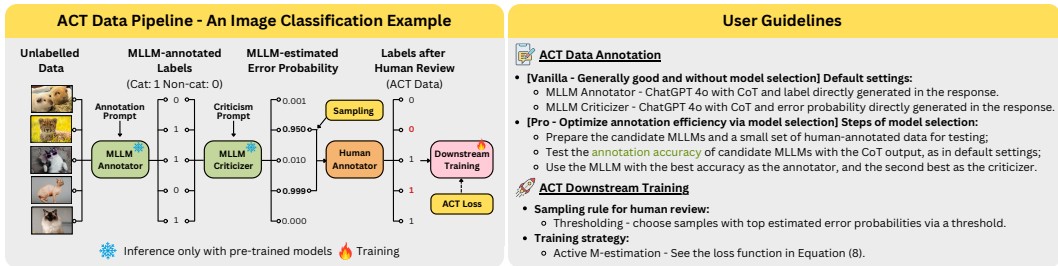

Figure 1: Illustration of the ACT data pipeline (left) and the user guidelines (right).

(MLLMs), thus supporting a wider range of practical domains, such as computer vision (CV) and multimodal understanding. In addition, unlike some existing methods that are limited to white-box models (i.e., those with accessible internal representations such as logit scores) [18, 19], our approach is agnostic to model accessibility. This allows us to effectively unify both white-box and black-box MLLMs, and enables the exploration into a broader spectrum of model families. Furthermore, in contrast to some related works [16, 17, 19], our data pipeline is training-free, waiving the need for additional training costs on annotators or criticizers.

The primary goal of this paper is to introduce a practically valuable and user-friendly data annotation pipeline. Specifically, we aim to enable future users—when faced with a new data annotation task—to quickly determine how to set up the entire pipeline and seamlessly deploy it in production. To this end, we preface a summary of key takeaways in the form of user guidelines in Figure 1, which distill the core insights from subsequent explorations in our paper into actionable steps.

The main discussions in this paper are divided into two parts, each addressing an important aspect of the annotation and training process. The first part (Sections 2 to 4) focuses on the annotation stage, where we explore how to choose appropriate MLLMs and prompt strategies to enhance label accuracy while efficiently reducing human effort. The second part (Section 5) addresses the challenge of downstream training. We theoretically analyze how to optimally sample data for human review and appropriately modify the loss function, such that models trained on ACT data achieve similar performance to those trained on fully human-annotated datasets. Empirically, we validate that, using the loss modified from active M-estimation [20], the performance gap can be narrowed to within 2% while saving up to 90% of human costs.

## 2 ACT Data Pipeline: Formulation & Key Evaluation Metrics

In this section, we provide a formal mathematical formulation of the ACT data pipeline and introduce key metrics to evaluate its performance.

**Data Annotation.** Let $\mathcal{X}$ be a space of unlabeled data with a finite space of labels $\mathcal{Y}$. We then denote the dataset as $\mathcal{D} = \{(\mathbf{x}_i, y_i)|\mathbf{x}_i \in \mathcal{X}, y_i \in \mathcal{Y}, i \in \mathcal{I}\}$, where the ground truth label $y_i$ is assumed to be unknown, and $\mathcal{I} = \{1, 2, ..., N\}$ is a set of indices. We formulate any data annotation approach as a function from the functional space $\mathcal{F} = \{f : \mathcal{X} \to \mathcal{Y}\}$ that maps the unlabeled data to labels. Ideally, given a dataset $\mathcal{D}$, the optimal data annotation $f^* \in \mathcal{F}$ satisfies $f^*(\mathbf{x}_i) = y_i, \forall i \in \mathcal{I}$. We further denote the sets of all annotations conducted by humans and machines as $\mathcal{F}^{(h)} \subset \mathcal{F}$ and $\mathcal{F}^{(m)} \subset \mathcal{F}$, respectively. Without loss of generality, we consider a representative human annotator $f^{(h)} \in \mathcal{F}^{(h)}$ and an MLLM-based machine annotator $f^{(m)} \in \mathcal{F}^{(m)}$ in following discussions. Let $\hat{y}_i^{(h)} = f^{(h)}(\mathbf{x}_i)$ and $\hat{y}_i^{(m)} = f^{(m)}(\mathbf{x}_i)$ be the human- and machine-annotated labels, respectively.

**Measure Annotation Quality.** We define a scalar function $Q : \mathcal{F} \times \mathcal{F} \to [Q_{\min}, Q_{\max}]$ as the quality measure. Formally, we evaluate the quality of an annotation method $f \in \mathcal{F}$ as:

$$Q(f^*, f) = \mathbb{E}_{\mathbf{x} \in \mathcal{X}}[s(f^*(\mathbf{x}), f(\mathbf{x}))] \approx \frac{1}{N} \sum_{i \in \mathcal{I}} s(f^*(\mathbf{x}_i), f(\mathbf{x}_i)), \qquad (1)$$

where $s(\cdot, \cdot)$ is a similarity function defined on the label space, such as 0-1 loss when $Q$ is accuracy, thus the higher value of $Q$ indicates better quality. We generally observe that $Q(f^*, f^{(h)}) > Q(f^*, f^{(m)})$, indicating that human annotation is usually more accurate than the machine. Following the common practice of supervised learning, where human-annotated labels are used as the ground

truth labels when the latter is not available, we assume that $\hat{y}_i^{(h)} = y_i$ for most if not all $i \in \mathcal{I}$, thus $Q(f^*, f^{(h)}) \approx Q(f^*, f^*) = Q_{\max}$.

**Review Machine Errors with Human.** Assume that $\delta \in \{0, 1\}^N$ is a machine error indicator, where $\delta_i = 1$ means $\hat{y}_i^{(m)} \neq y_i$, and 0 otherwise. Then, the refinement of machine annotation can be done via a correction operator: $\kappa_\delta[f^{(m)}](\mathbf{x}_i) = (1 - \delta_i) f^{(m)}(\mathbf{x}_i) + \delta_i f^*(\mathbf{x}_i)$. Practically, regarding human-annotated labels as the ground truth, the correction operator can be approximated by the human-correction operator: $\kappa_\delta[f^{(m)}](\mathbf{x}_i) \approx \kappa_\delta^{(h)}[f^{(m)}](\mathbf{x}_i) = (1 - \delta_i) f^{(m)}(\mathbf{x}_i) + \delta_i f^{(h)}(\mathbf{x}_i)$. Accordingly, our goal is to find $\delta$ for the following optimization problem:

$$\max_\delta Q\left(f^{(h)}, \kappa_\delta^{(h)}[f^{(m)}]\right). \tag{2}$$

Ideally, the optimal solution is the Kronecker delta $\delta_{\hat{y}^{(m)}, \hat{y}^{(h)}}$ such that $\delta_{\hat{y}_i^{(m)}, \hat{y}_i^{(h)}} = 1$ if $\hat{y}_i^{(m)} \neq \hat{y}_i^{(h)}$, and 0 otherwise. This means we replace the erroneous machine-annotated labels with the human-annotated labels.

**ACT Data Pipeline.** Now the problem is, we do not know when the machine makes mistakes. Previous works have validated the judgmental capabilities of LLMs, where an LLM evaluates answers generated by another LLM or itself to enhance the generator's inference quality [21, 22, 23, 24]. Enlightened by this idea, we adopt a pre-trained MLLM $g$ as the criticizer to determine whether the annotator $f^{(m)}$ assigns the wrong label for each data pair $(\mathbf{x}_i, \hat{y}_i^{(m)})$. Considering that human resources are limited, instead of generating binary decisions, we query the criticizer to estimate the probability of error. Then, we sample the data for human review based on this estimated probability, ensuring that the total sample size does not exceed a given human budget $B$. Mathematically, let $\epsilon_i = \mathbb{P}(y_i \neq \hat{y}_i^{(m)} | \mathbf{x}_i)$ denote the true error probability, which is then estimated by $\hat{\epsilon}_i = g(\mathbf{x}_i, \hat{y}_i^{(m)})$. Recall that $N$ is the dataset size. Given a human budget $B \leq N$, we define a budget-aware sampling function $\delta(B)$ where $\delta_i(B) \sim \mathbb{B}(\pi_B(\hat{\epsilon}_i))$. Here, $\mathbb{B}$ denotes the Bernoulli distribution, and $\pi_B(\cdot)$ is a transformation that adjusts the error probability $\hat{\epsilon}_i$ based on the budget $B$, ensuring that $\sum_\mathcal{I} \delta_i(B) \leq B$. In this paper, we consider the following sampling rules:

- *Normalization* [17, 20]: $\pi_B(\hat{\epsilon}_i) = (B \times \hat{\epsilon}_i) / \sum_\mathcal{I} \hat{\epsilon}_i$ such that $\sum_\mathcal{I} \delta_i(B) \leq B$.

- *Exponential Weighting*: $\pi_B(\hat{\epsilon}_i) = 1 / (1 + \mathrm{e}^{-\beta(\hat{\epsilon}_i - \alpha)})$, where $\alpha \in [0, 1]$ and $\beta \in \mathbb{R}_+$ are hyperparameters that control the center and sharpness of the transformed error distribution. The hyperparameters are set to satisfy $\sum_\mathcal{I} \delta_i(B) \leq B$.

- *Thresholding*: $\pi_B(\hat{\epsilon}_i) = \mathbf{1}(\hat{\epsilon}_i \geq \tau)$, where $\tau$ is a sampling threshold chosen such that $\sum_\mathcal{I} \delta_i(B) \leq B$. This is a special case of exponential weighting when $\alpha = \tau$ and $\beta \to \infty$.

With the human budget $B$, our ultimate goal is modified from Equation (2) by adding the budget constraint as

$$\max_{\delta(B)} Q\left(f^{(h)}, \kappa_{\delta(B)}^{(h)}[f^{(m)}]\right), \quad \text{s.t.} \sum_{i \in \mathcal{I}} \delta_i(B) \leq B, \tag{3}$$

where $\kappa_{\delta(B)}^{(h)}[f^{(m)}](\mathbf{x}_i) = (1 - \delta_i(B)) f^{(m)}(\mathbf{x}_i) + \delta_i(B) f^{(h)}(\mathbf{x}_i)$ is the budget-constrained human correction operator.

As a summary, the ACT data pipeline consists of three steps: (1)Annotation: generating MLLM-annotated labels for all unlabeled data in the dataset; (2) Error estimation: utilizing another MLLM as the criticizer to estimate the error probability for each annotation job; (3) Correction: sample data with high likelihood of machine annotation errors for human review and correction, with the sample size controlled by a human budget.

**Metrics.** In the following sections, we will evaluate the effectiveness of various MLLMs and prompt strategies as components of the ACT data pipeline. Here, we propose two key metrics: (1) Annotation Quality Gain (AQG): measurement of the annotation quality improved by ACT from machine annotation given a fixed human budget $B$; (2) Area under Budget Sensitivity (ABS): measurement of the overall efficiency of human budget, for which a higher value implies better budget utilization.

**Definition 2.1 (Annotation Quality Gain).** *With a machine annotator $f^{(m)}$, a machine criticizer $g$, and a correction operator $\kappa_{\delta(B)}^{(h)}$ under human budget $B$, where $\delta_i(B) \sim \mathbb{B}(\pi_B(\hat{\epsilon}_i))$ with $\hat{\epsilon}_i =$*

$g(\mathbf{x}_i, f^{(m)}(\mathbf{x}_i))$, we define the Annotation Quality Gain (AQG) of the data pipeline as

$$\text{AQG}\left(f^{(m)}, g, B\right) = \frac{Q(f^{(h)}, \kappa_{\delta(B)}^{(h)}[f^{(m)}]) - Q\left(f^{(h)}, f^{(m)}\right)}{Q_{\max} - Q\left(f^{(h)}, f^{(m)}\right)}, \tag{4}$$

where the denominator $Q_{\max} - Q(f^{(h)}, f^{(m)})$ scales AQG into range $[0, 1]$, which offsets the influence of room for improvement.

**Definition 2.2 (Area under Budget Sensitivity).** *Let $b \in [0, 1]$ be the proportion of human budget over the dataset, such that $B = \lfloor bN \rfloor$, where $\lfloor \cdot \rfloor$ means round down to the largest integer. With a machine annotator $f^{(m)}$ and a machine criticizer $g$, we define the Area under Budget Sensitivity (ABS) of the data pipeline as*

$$\text{ABS}\left(f^{(m)}, g\right) = \int_0^1 \text{AQG}\left(f^{(m)}, g, \lfloor bN \rfloor\right) db \approx \frac{1}{N} \sum_{B=0}^{N} \text{AQG}\left(f^{(m)}, g, B\right). \tag{5}$$

## 3 Experimental Details

**Datasets.** Since MLLMs are capable of handling multiple data types, we consider a comprehensive variety of annotation tasks, including image classification: CIFAR10 [25], Fashion-MNIST (Fashion) [26], and Stanford Cars (Cars) [27]; text classification: Emotion [28] and Irony [28]; and vision question-answering (VQA): the close-ended VQA-RAD [29].

**MLLMs.** We conduct experiments using models from six prominent MLLM families: Chat-GPT 4o 2024-08-06 (GPT4o) [30], Gemini-1.5-Pro-002 (Gemini1.5P) [31], Claude 3.5 Sonnet (Claude3.5S) [32], LLaVA OneVision 72B (LLaVA-OV) [33], Qwen 2.5 VL 72B (Qwen2.5VL) [34], and InternVL 2.5 78B (InternVL2.5) [35]. Among these models, LLaVA-OV, Qwen2.5VL, and InternVL2.5 are white-box models that provide access to intermediate outputs such as logit scores, facilitating further fine-grained analyses. In contrast, GPT4o, Gemini1.5P, and Claude3.5S are black-box models, offering only final responses. Larger models are generally associated with better criticism functionality [24], hence, we select white-box models with large parameter sizes - each exceeding 70 billion, to evaluate the data pipeline. For all models, we adopt sampling hyperparameters $p = 0.9, t = 0.7, k = 50$, and max new tokens 500 when applicable.

**Downstream Models.** We employ three pre-trained models for our downstream experiments. For image classification tasks, we use ResNet18 [36], initialized with default weights pretrained on ImageNet-1k [37]. For text classification, we utilize the RoBERTa-base model (RoBERTa) [38]. For the VQA task, we adopt the BLIP-VQA model (`blip-vqa-base`) [39].

More experimental details of the datasets and downstream training can be found in Appendix A.

## 4 What Makes ACT Work Better?

In this section, we investigate how to boost ACT's performance by systematically exploring each component, namely, the machine annotator, the criticizer, and the associated prompt strategies. We choose accuracy as the annotation quality measure. In addition, we do not assume a fixed human annotation budget and adopt ABS as the evaluation metric to capture the pipeline's overall efficiency in leveraging human resources to boost the annotation quality. In consideration of time and computational costs, results reported are from a single run per experiment; but a stability analysis in Appendix B demonstrates the robustness of the observations. Examples of prompt strategies explored are provided in Appendix C.

### 4.1 Exploration of Machine Annotators

We consider two types of prompt strategies for machine annotators: (1) Naïve annotation (naïve): directly output the label; (2) Annotation with Chain-of-Thought (CoT) [40]: output the step-by-step reasoning with the label. The results are shown in Figure 2.

**Insight 1: GPT4o is a generally promising annotator.** We observe that GPT4o achieves the highest annotation accuracy on most datasets. While LLaVA-OV performs best on VQA-RAD, it struggles with image classification tasks (CIFAR-10 and Fashion).

**Insight 2: CoT is not consistently helpful with annotation.** The results do not show a clear benefit from using CoT overall. However, for GPT4o specifically, CoT improves performance across all

tasks. Therefore, in the following exploration of criticizers, we will adopt GPT4o with the CoT prompting strategy as the base annotator.

## 4.2 Exploration of Machine Criticizers

Recall that we categorize MLLMs into two types: black-box and white-box, based on whether they provide access to the underlying logit probabilities. Accordingly, we design several strategies for each type to measure the error probabilities of the annotator. For simplicity, we use the thresholding sampling rule in the criticizer experiments. In Appendix B, we demonstrate that different sampling rules have minimal impact on the ABS results.

### 4.2.1 Black-box Criticism Strategies

Black-box criticism strategies refer to methods that query models for error elicitation in the response token space. These strategies are broadly applicable to any model with chatbot functions, which means they can also be applied to white-box models. In our study, we evaluate four black-box strategies. While they may not encompass every possible approach, they are representative and straightforward to implement. A detailed description of these strategies is provided in Table 1. Since the mc strategy implicitly measures error probability through error levels, we use threshold-based sampling with random selection when multiple options fit the budget. Other sampling rules may not directly apply to this strategy. Note that the mc and devil strategies are CoT variants differing mainly in their error measurement approach. The results of the black-box strategies are presented in Figure 3.

Table 1: Details of black-box criticism strategies.

| Strategy | Description | Input (Prompt) → Output (Response) |
|---|---|---|
| Naïve Estimation (naïve) | Directly estimate the error probability based on the given data and label. | `data` & `label` → `error_prob` |
| Chain-of-Thought (CoT) [40] | Output step-by-step reasoning and error probability. | `data` & `label` → `CoT` & `error_prob` |
| Multiple Choice (mc) [41, 42, 43] | Select from predefined error levels from 1 to 5, where higher level means higher error probability. | `data` & `label` → `CoT` & `error_level` |
| Devil's Advocate (devil) [44, 45] | Critically assess the CoT produced by the annotator and estimate its error probability. | `data` & `CoT_A` → `CoT` & `error_prob` |

*`CoT_A` is the CoT obtained from the machine annotator.

**Insight 3: Cross-criticism outperforms self-criticism with black-box strategies.** Recall that we use GPT4o as the machine annotator. When GPT4o also serves as the criticizer, the process is referred to as self-criticism. In contrast, when the criticizer is another model, the process is known as cross-criticism. Figure 3 indicates that the best ABS scores are typically achieved through cross-criticism. However, the performance of self-criticism remains highly competitive in most datasets.

**Insight 4: Black-box models are better criticizers with black-box strategies.** We generally observe that black-box models (GPT4o, Gemini1.5P, & Claude3.5S) achieve higher ABS scores than white-box models (LLaVA-OV, Qwen2.5VL, & InternVL2.5) when using black-box strategies. Specifically, GPT4o and Gemini1.5P tend to perform better on image classification tasks, while Claude3.5S demonstrates stronger performance on language-focused tasks, including text classification and VQA.

**Insight 5: CoT is more helpful with criticism than annotation.** Compared to our observations with the machine annotator, CoT consistently shows more benefits in the criticism process, leading to an ABS improvement of up to 22.46% over the naïve strategy. We suppose that this is because

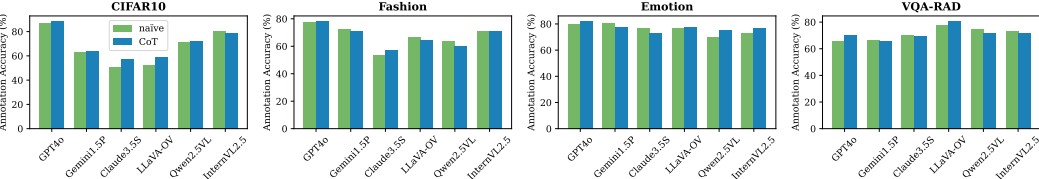

Figure 2: Accuracy of various models as machine annotators with different prompt strategies.

detecting errors requires explicit reasoning and justification, whereas annotation often relies more on direct pattern recognition. CoT also demonstrates a clear advantage over other prompt strategies, achieving the best results on 4 out of the 6 datasets.

#### 4.2.2 White-box Criticism Strategies

With white-box models, we propose two approaches to estimate the error probability: logit probability and perplexity (PPL). In the logit probability approach, we ask the criticizer to predict if there is an error and answer `"yes"` or `"no"`. The estimated error probability is then calculated as:

$$\hat{\epsilon} = \mathbb{P}(\texttt{"yes"})/(\mathbb{P}(\texttt{"yes"}) + \mathbb{P}(\texttt{"no"})), \qquad (6)$$

where $\mathbb{P}(\texttt{"yes"})$ and $\mathbb{P}(\texttt{"no"})$ denote the logit probabilities assigned to tokens `"yes"` and `"no"`. In the PPL-based approach, we adopt the CoT prompt strategy and indirectly measure the error by computing the PPL of the criticizer's CoT. We note that only the threshold-based sampling is applicable for PPL, as we cannot directly estimate the error probability. During sampling, instances are prioritized based on the following selection order:

$$\downarrow \text{PPL}(\boxed{\text{CoT}}|\texttt{"yes"}) \succ \uparrow \text{PPL}(\boxed{\text{CoT}}|\texttt{"yes"}) \succ \uparrow \text{PPL}(\boxed{\text{CoT}}|\texttt{"no"}) \succ \downarrow \text{PPL}(\boxed{\text{CoT}}|\texttt{"no"}), \quad (7)$$

where $\text{PPL}(\boxed{\text{CoT}}|\texttt{"yes"})$ and $\text{PPL}(\boxed{\text{CoT}}|\texttt{"no"})$ are corresponding to the PPL of the CoT when the final decision is `"yes"` and `"no"`; $\uparrow$ and $\downarrow$ means higher and lower PPL values; and $\succ$ indicates higher priority in the sampling process. For further clarity, we summarize the white-box strategies in Table 2. The comparison between the black- and white-box strategies are shown in Figure 4.

Table 2: Details of white-box criticism strategies.

| Strategy | Description | Input (Prompt) → Output (Response) → Error Measurement |
|---|---|---|
| **Naïve Estimation with Logit Probability (naïve-logit) [46, 47, 48]** | Output `"yes"` or `"no"` based on the given data and label. Error probability is estimated by the logit probability. | `data` & `label` → `"yes"`/`"no"` 
 → Error estimation via logit probabilities |
| **Chain-of-Thought with Logit Probability (CoT-logit)** | Output step-by-step reasoning before `"yes"` or `"no"`. Error probability is estimated by the logit probability. | `data` & `label` → `CoT` & `"yes"`/`"no"` 
 → Error estimation via logit probabilities |
| **Chain-of-Thought with Perplexity (CoT-PPL)** | Output step-by-step reasoning before `"yes"` or `"no"`. Error is indirectly measured by PPL of the CoT. | `data` & `label` → `CoT` & `"yes"`/`"no"` 
 → PPL calculation |

**Insight 6: White-box strategies can occasionally enhance criticism performance.** Compared to black-box strategies, models often achieve higher ABS scores with white-box methods, particularly with naïve-logit and CoT-PPL. This improvement is more consistently observed with InternVL2.5, while LLaVA-OV and Qwen2.5VL show smaller gains.

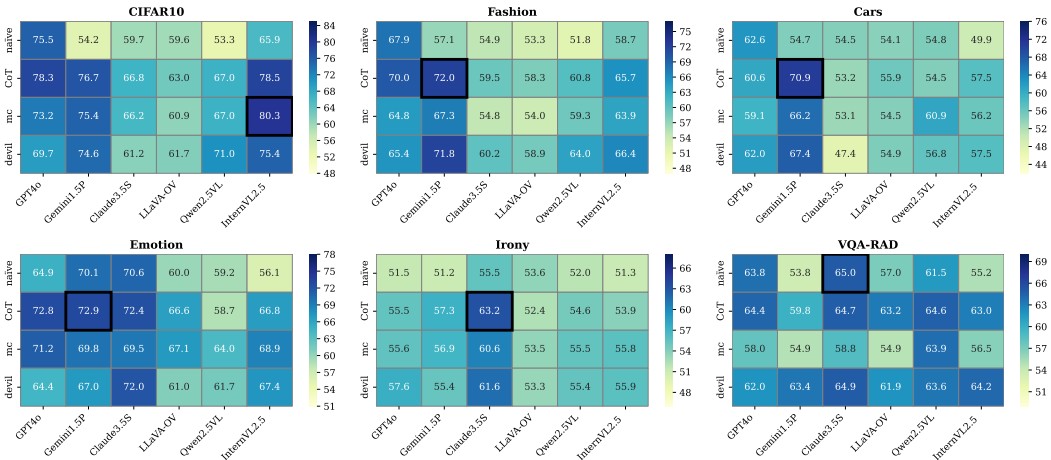

Figure 3: Results of black-box strategies. The metric shown is ABS (%), where higher values indicate better annotation efficiency of the ACT data pipeline. The best results are highlighted with black frames.

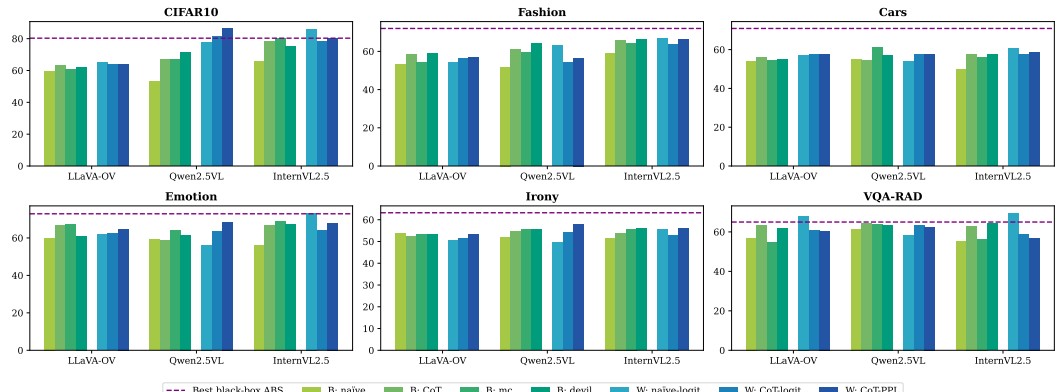

Figure 4: Comparison of ABS (%) for the same model using black-box and white-box strategies. Green bars correspond to black-box strategies (B), while blue bars represent white-box strategies (W). The horizontal dashed lines indicate the best black-box results for reference.

**Insight 7: White-box strategies do not consistently outperform black-box strategies.** Although white-box methods occasionally outperform the best black-box ABS, this occurs in only 2 of the 6 datasets tested. The inconsistent performance arises primarily because white-box strategies cannot leverage powerful black-box models, limiting their effectiveness on the same tasks. Therefore, we conclude that black-box strategies remain more promising for practical use based on current results. However, we also conjecture that, as MLLMs continue to advance, a future white-box model with capabilities comparable to black-box models could make it possible for white-box strategies to consistently achieve superior performance.

## 4.3 Bridging Insights and Practical Usability of the ACT Data Pipeline

From the explorations above, we have gained some insights that lead to practical guidelines: (1) GPT4o with the CoT strategy can serve as a generally good annotator; (2) Black-box criticism strategies, especially the CoT strategy, provide more generalized practical values with current MLLMs; (3) While GPT4o self-criticism is a promising setting for the ACT data pipeline, the best ABS is typically achieved through cross-criticism. Building on these, we further examine model selection to offer guidance that makes ACT more user-friendly.

The key question of interest is: for a given dataset, how do we choose the annotator and criticizer to maximize pipeline efficiency? Of course, the effortless default is GPT4o self-criticism with CoT, which often works well but may not be optimal. To facilitate better selection, we propose a simple but effective approach based on empirical observations (Figure 5). First, annotation and criticism abilities are positively correlated, implying that models with strong annotation ability are also likely to perform well as criticizers. Second, across datasets, the best criticism ABS are consistently achieved by using the top-performing annotator and the model with second-best annotation ability (i.e., best among models other than the annotator) as criticizer. We therefore recommend: (1) obtaining a small test set of human-annotated samples; (2) evaluating annotation ability across available MLLMs; and (3) selecting the best model as the annotator and the second-best as the criticizer.

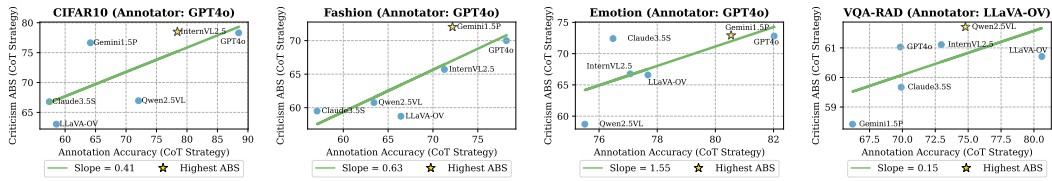

Figure 5: Relationship between annotation and criticism abilities. The base annotators for criticism ability evaluation are shown in the titles (all with top 1 annotation ability from Figure 2). Each point represents a model. The models with the highest ABS are highlighted with stars. The slopes of the fitted regression lines are indicated in the legends.

# 5 ACT as Human: Enhance Downstream Performance with Modified Loss

## 5.1 Theoretical Discussions of Downstream Training with ACT Data

**Problem: The Downstream Challenge.** With a dataset annotated by the ACT data pipeline (referred to as ACT data), the next challenge lies in effectively leveraging it for downstream tasks, such as supervised fine-tuning (SFT). The ACT data consists of two components: the high-quality human-annotated data, and the comparatively lower-quality machine-annotated data, which may include erroneous labels. Relying solely on human-annotated data results in significant data loss, particularly when human budget is limited. On the other hand, incorporating both human- and machine-annotated data risks introducing noisy labels, which may disturb the learning process and lead to suboptimal performance in downstream tasks.

**Solution: Active M-estimation.** A potential solution to the downstream challenge is active M-estimation, which proposed a modified loss to ensure effective statistical properties that lead to promising downstream training [17, 20]. In this work, we adapt the modified loss to ACT data and design the ACT loss as follows:

$$\mathcal{L}_\theta^{(ACT)} = \frac{1}{N} \sum_{i=1}^{N} \left( \ell_{\theta,i}^{(m)} + \left( \ell_{\theta,i} - \ell_{\theta,i}^{(m)} \right) \frac{\delta_i(B)}{\pi_B(\hat{\epsilon}_i)} \right), \tag{8}$$

where $\ell_\theta$ and $\ell_\theta^{(m)}$ are losses computed with ground truth and machine-annotated data, respectively. $\delta_i(B) \sim \mathbb{B}(\pi_B(\hat{\epsilon}_i))$ is the error indicator as discussed in Section 2. In practical implementation, ground-truth loss is usually estimated with the high-quality human-annotated data.

According to Proposition 5.1 (see proof in Appendix D), while the ACT loss is an unbiased estimate of the ground truth loss, we note that the variance of ACT loss is minimized when the expectation term in Equation (9) equals to 0. This can be satisfied or mostly satisfied with: (1) *perfect machine annotator*: the machine annotator is perfectly accurate such that $\ell_\theta^{(m)} = \ell_\theta^{(h)} \approx \ell_\theta$; (2) *astute machine criticizer:* when the machine annotation is not sufficiently accurate, the second term can be reduced by accurate error probability estimation, i.e., $\pi_B(\hat{\epsilon}) \to 1$ when machine makes mistakes. This further demonstrates the importance of our exploration in Section 4.

> **Proposition 5.1 Statistical Properties of ACT Loss** »
>
> *Suppose $\mathcal{L}_\theta = \mathbb{E}(\ell_\theta(\mathbf{x}, y))$ is a $\mu$-strongly convex loss. Let $\ell_\theta^{(m)} = \ell_\theta(\mathbf{x}, \hat{y}^{(m)})$ be the loss computed with machine-annotated data. Then, the ACT loss defined in Equation (8) is an unbiased loss estimate: $\mathbb{E}\left( \mathcal{L}_\theta^{(ACT)} \right) = \mathcal{L}_\theta$ [20]. The variance of the ACT loss is given by:*
>
> $$\mathrm{Var}\left( \mathcal{L}_\theta^{(ACT)} \right) = \frac{1}{N} \left( \mathrm{Var}\left( \ell_\theta \right) + \mathbb{E}\left[ \left( \ell_\theta - \ell_\theta^{(m)} \right)^2 \left( \frac{1}{\pi_B(\hat{\epsilon})} - 1 \right) \right] \right). \tag{9}$$

**Important Factor: The Sampling Rule.** In Section 2, we introduced three sampling rules to keep the volume of data for human review within a pre-specified human budget. While all rules adapt to the ACT loss and share similar statistical properties, they can yield very different training outcomes. We show via Theorem 5.2 (see proof in Appendix D) that when the machine annotator is fixed (i.e., $C$ is fixed), the upper bound of the gap between parameters learned with ACT loss and ground truth loss is decided by $q$, the lower bound of the transformed errors of samples reviewed by human. A small $q$ increases the likelihood of a large parameter gap and poor model performance. Active M-estimation [17, 20] uses the normalization rule, but this may perform poorly when the human budget $B$ is small. This is because $\pi_B(\hat{\epsilon}_i) = (B \times \hat{\epsilon}_i)/\sum_{\mathcal{I}} \hat{\epsilon}_i$ becomes close to 0 ($q \to 0$) when $B$ is small compared to the accumulation of errors in a large dataset. To address this, we propose exponential weighting and thresholding rules, which map errors of selected samples close to 1 ($q \to 1$), ensuring a small parameter gap, and eventually, model performance similar to the ground truth loss. Further details of the ACT losses with different sampling rules are presented in Appendix E.

**Theorem 5.2 Probabilistic Upper Bound of the Parameter Gap**  ≫

*Let $N$ be the dataset size. Suppose $\ell_\theta$ is $\mu$-strongly convex, and there exist constants $q, C > 0$ such that the transformed error $\pi_B(\hat{\epsilon}_i) \geq q$ for all $i$ with $\delta_i(B) = 1$, and the gradient gap $\|\nabla \ell_{\theta,i}^{(m)} - \nabla \ell_{\theta,i}\| \leq C$ for all $i \in \{1, 2, ..., N\}$. Then, for an arbitrary $p \in (0,1)$, with probability at least $1 - p$, we can bound*

$$\|\theta_*^{(ACT)} - \theta_*\| \leq \sqrt{\frac{8c_1 \log(2/p)}{\mu^2 N}}, \tag{10}$$

*where $\theta_*^{(ACT)} = \arg\min_\theta \mathcal{L}_\theta^{(ACT)}$, $\theta_* = \arg\min_\theta \mathcal{L}_\theta$, and $c_1 = (1-q)C^2/q$.*

## 5.2 Experiments of Downstream Training with ACT Losses

To empirically support our theoretical analysis, we conducted experiments across various data and loss combinations, with results presented in Table 3. Without affecting the theoretical validity, we adopt the Cross-entropy loss as $\ell_\theta$ and apply the power-tuning hyperparameter for all ACT losses following [17]. We used GPT4o with the CoT strategy for both annotation and criticism as the default settings of ACT. The human budget is determined by GPT4o's annotation accuracy—for instance, with 88.48% accuracy on CIFAR10, the human budget is set to 11.52% of the dataset. We treat this as the "ideal" budget, as it is the least budget we shall assign to fix all machine errors. In practice, however, the human budget should be set based on available resources. To further enhance the completeness of our analyses, we provide the sensitivity analysis of human budget in Appendix F.

We highlight four key observations from the results. First, training solely on machine-annotated data yields up to 10.15% lower accuracy compared to human-annotated data, whereas ACT reduces the performance gap to less than 2% for most datasets while saving approximately $70\% \sim 90\%$ human costs. Second, exponential weighting (exp.) and thresholding (thre.) outperform normalization (norm.), especially under limited human budgets. On Cars, with an ideal budget of 9.56%, normalization leaves a 76.34% gap from full supervision, while exponential weighting and thresholding reduce it to 1.69% and 1.88%. Third, ACT data with Cross-entropy loss also perform comparably well, as it is a special case of thresholding (see Appendix E). Lastly, exponential weighting and thresholding perform similarly overall. Although exponential weighting often yields slightly better results, we recommend thresholding for its simplicity, as the threshold $\tau$ is easier to decide with a given human budget than the hyperparameters in the exponential weighting transform ($\alpha$ and $\beta$).

Table 3: Results of downstream training - test accuracy (%) in form of mean $\pm$ std over 5 runs.

| Training Data - Loss | CIFAR10 (ResNet18) | Fashion (ResNet18) | Cars (ResNet18) | Emotion (RoBERTa) | Irony (RoBERTa) | VQA-RAD (BLIP-VQA) |
|---|---|---|---|---|---|---|
| Human only - Cross-entropy Loss | $88.66 \pm 0.97$ | $93.01 \pm 0.63$ | $87.88 \pm 0.36$ | $81.82 \pm 0.57$ | $70.18 \pm 3.23$ | $67.81 \pm 1.47$ |
| Machine only - Cross-entropy Loss | $81.55 \pm 1.93$ | $82.86 \pm 0.84$ | $83.68 \pm 0.17$ | $78.96 \pm 2.40$ | $60.71 \pm 5.43$ | $61.03 \pm 2.05$ |
| ACT data - Cross-entropy loss | $85.59 \pm 0.52$ | $87.50 \pm 0.86$ | $85.88 \pm 0.26$ | $80.82 \pm 1.08$ | $67.83 \pm 2.82$ | $61.83 \pm 3.27$ |
| ACT data - ACT norm. loss | $64.70 \pm 5.46$ | $69.27 \pm 7.25$ | $11.54 \pm 0.96$ | $79.87 \pm 0.88$ | $65.66 \pm 2.00$ | $62.55 \pm 3.01$ |
| **ACT data - ACT exp. loss (Ours)** | $87.73 \pm 0.36$ | $89.73 \pm 0.35$ | $86.19 \pm 0.14$ | $81.44 \pm 0.51$ | $68.49 \pm 3.20$ | $67.73 \pm 1.33$ |
| **ACT data - ACT thre. loss (Ours)** | $87.95 \pm 0.35$ | $89.16 \pm 0.89$ | $86.00 \pm 0.26$ | $81.41 \pm 0.64$ | $68.21 \pm 1.94$ | $67.02 \pm 1.32$ |
| Human-Machine performance gap (%) | 7.11% | 10.15% | 4.20% | 2.86% | 9.47% | 6.87% |
| Human-ACT performance gap (%) | 0.71% | 3.28% | 1.69% | 0.38% | 1.69% | 0.08% |
| ACT human budget (%) | 11.52% | 21.81% | 9.56% | 17.98% | 33.79% | 30.15% |

# 6 Related Works

**Data Annotation with LLMs** LLMs' annotation ability has been widely studied, but mainly in NLP [6, 7, 12, 13]. The most relevant to our work is CDI [17], which uses a trained XGBoost [49] to detect LLM errors for human review. However, CDI (1) employs an error detector that lacks flexibility, as it often requires task-specific design and training with additional data; and (2) uses normalization-based active M-estimation loss, which we find suboptimal in downstream experiments. We further discuss CDI and other prior works [15, 16, 18, 19], along with experimental comparisons, in Appendix G.

**LLM-as-a-Judge** LLM-as-a-Judge refers to using an LLM to evaluate the outputs generated by other LLMs [23, 50, 51, 52]. Many prior works considered the case where the verifier and generator are the same model, and focused on the phenomenon of LLM self-improvement [21, 22, 24, 53, 54, 55, 56], which is the concept we refer to as self-criticism in our study. However, recent studies suggest that

self-criticism may introduce bias, as the verifier tends to favor its own outputs [57, 58]. This may explain why we observe better performance with cross-criticism in our ACT data pipeline.

# 7 Conclusion

In this paper, we introduce the ACT data pipeline, which uses MLLM-generated annotations for most data while strategically allocating human effort to potentially erroneous cases, as identified by another MLLM criticizer. We further analyze how to modify the training loss to align the performance of models trained on ACT data with those trained on fully human-annotated data, supported by both theoretical analysis and empirical evidence. Although our study is based on current MLLMs, and the efficiency of the proposed pipeline is constrained by their capabilities, our approach can be readily adapted to more advanced models in the future. The insights gained from our explorations are likely to generalize and provide valuable guidance for practical applications and future research.

The main limitation of this work is that we focus primarily on classification tasks, without involving more complex tasks such as text summarization or open-ended question answering. We provide a discussion in the Appendix G on how our approach can be extended to these more complex scenarios.

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

# Appendices Contents

# A Supplementary Experimental Details

## A.1 Datasets

Table 4: Dataset details.

| Dataset | Type | Description | #Classes | Size - Train | Size - Test |
|---------|------|-------------|----------|--------------|-------------|
| CIFAR10 | CV | Image classification of basic image categories. | 10 | 50,000 | 10,000 |
| Fashion-MNIST | CV | Image classification of cloth items. | 10 | 60,000 | 10,000 |
| Stanford Cars | CV | Image classification of car models. | 196 | 8,143 | 8,040 |
| TweetEval-Emotion | NLP | Text emotion classification. | 4 | 3,257 | 1,421 |
| TweetEval-Irony | NLP | Text irony detection. | 2 | 2,862 | 784 |
| VQA-RAD (Close-end) | VQA | Question-answer pairs on radiology images. | 2 | 940 | 262 |

Table 5: Dataset sources and licenses retrieved from `https://paperswithcode.com/datasets`.

| Dataset | Source | License |
|---------|--------|---------|
| CIFAR10 | `https://www.cs.toronto.edu/~kriz/cifar.html` | `N/A` |
| Fashion-MNIST | `https://github.com/zalandoresearch/fashion-mnist` | `MIT` |
| Stanford Cars | `https://paperswithcode.com/dataset/stanford-cars` | `Custom (non-commercial)` |
| TweetEval-Emotion | `https://github.com/cardiffnlp/tweeteval` | `N/A` |
| TweetEval-Irony | `https://github.com/cardiffnlp/tweeteval` | `N/A` |
| VQA-RAD | `https://paperswithcode.com/dataset/vqa-rad` | `CC0 1.0 Universal` |

## A.2 Device & Random Seed

All experiments are conducted with 1 to 8 NVIDIA SXM5 H100 GPUs with 80GB memories. When applicable, we set the random seed to 42 for all controllable sources of randomness.

## A.3 Downstream Training

**Sampling** For the thresholding sampling rule, the threshold $\tau$ is determined by the quartile corresponding to the human budget proportion. Specifically, we rank the errors in descending order and set $\tau$ as the quartile value that matches the given proportion of the human budget. For the exponential weighting sampling rule, we try $\beta = 10$ or $100$ for all datasets. The mean transition parameter $\alpha$ is set in the same way as $\tau$, based on the corresponding quartile.

**ResNet18** For all datasets, we fine-tune the ResNet18 model initialized with ImageNet-pretrained weights for 10 epochs. The batch size is set to 4096 for CIFAR-10 and Fashion-MNIST, and 32 for the Stanford Cars dataset. We use the Adam optimizer for CIFAR-10 and Fashion-MNIST, while the SGD optimizer is employed for Cars, following the implementation described in `https://www.kaggle.com/code/archanatrivedi/resnet18-on-stanford-car-dataset`. The key hyperparameters include the learning rate, with a search space of [$1e$-2, $1e$-3, $5e$-4, $1e$-4], and the power-tuning parameter for ACT losses, with values selected from [0.6, 0.7, 0.8, 0.9, 1.0] (see Appendix E for more details about the power-tuning parameter).

**RoBERTa** For text classification tasks, we fine-tune the RoBERTa-base model for 5 epochs. We set the batch size to 32 and use the AdamW optimizer. The key hyperparameters include the learning rate, with a search space of [$1e$-4, $5e$-5, $2e$-5, $1e$-5], and the power-tuning parameter for ACT losses, with values selected from [0.6, 0.7, 0.8, 0.9, 1.0].

**BLIP-VQA** For VQA-RAD, we fine-tune the BLIP-VQA model initialized with the `blip-vqa-base` for 10 epochs. Since we adopt the close-ended version of VQA-RAD, where answers are limited to either "Yes" or "No", we use these two tokens as ground truth. During both training and inference, the model is prompted to generate either "Yes" or "No" in the response token space. We evaluate the trained model by checking if the first generated response token—either 'Yes' or 'No'—matches the correct answer. We use a batch size of 32 and optimize the model using the AdamW optimizer. The key hyperparameters include the learning rate, searched over [$2e$-4, $1e$-4, $5e$-5, $2e$-5], and the power-tuning parameter for ACT losses, selected from [0.6, 0.7, 0.8, 0.9, 1.0].

# B Stability Analyses

## B.1 Robustness of Annotation and Criticism Results

We assess the robustness of both annotation and criticism by repeating the process 5 times for each MLLM involved in our explorations. We perform the stability test on a subset of all datasets (i.e., 100 random samples per class). The results are shown in Figure 6, Figure 7, and Figure 8, respectively. We observe that the standard deviations are generally low (most within 2%). In addition, the rank of abilities does not change after taking account of potential variations in metric values, indicating that the conclusions drawn from our explorations are robust to randomness.

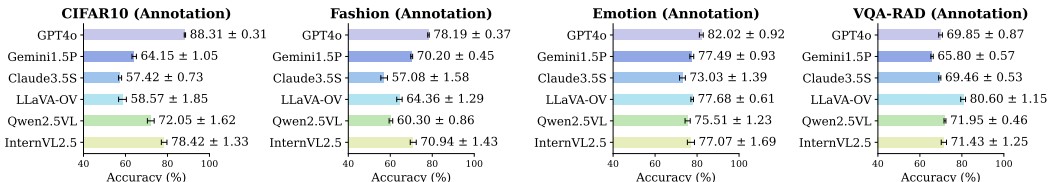

Figure 6: Annotation accuracy with error bars (mean ± std). Results are presented for the CoT prompt strategy across 6 MLLM annotators.

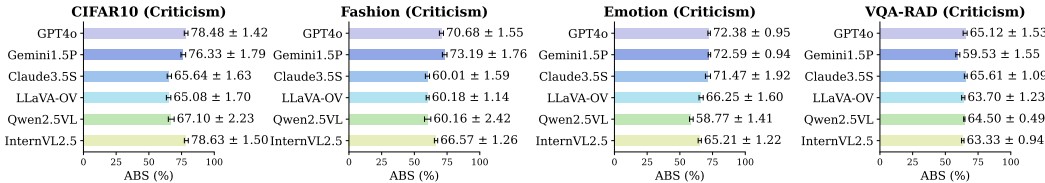

Figure 7: Criticism ABS with error bars (mean ± std). Results are presented for the black-box CoT prompt strategy across 6 MLLM criticizers.

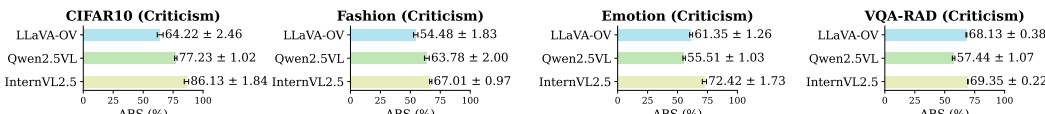

Figure 8: Criticism ABS with error bars (mean ± std). Results are presented for the white-box naïve-logit prompt strategy across 3 MLLM criticizers.

## B.2 Impact of Different Sampling Rules on Criticism

In Figure 9, we present results computed on the full datasets using different sampling rules to evaluate the criticism ABS. The results indicate that the choice of sampling rule has minimal impact on the comparative outcomes. Therefore, the insights derived from our analyses using the thresholding rule remain consistent across other sampling methods as well.

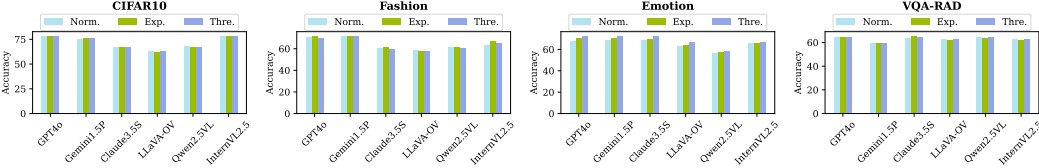

Figure 9: Criticism ABS with various sampling rules. Results are presented for the black-box CoT strategy across 6 MLLM criticizers.

# C   Examples of Prompt Strategies

Here, we present examples of prompt strategies. For annotation prompt - naïve, we include examples for all three tasks (image classification, text classification, and VQA) to provide a comprehensive illustration. For other prompt strategies, we only present examples for image classification, but they can be easily adapted to other tasks by following the same pattern as the annotation prompts. In the following examples, {purple} denotes inputs, while [blue] denotes outputs. Note that we require MLLMs to follow a specific output format to simplify the extraction of CoT and labels. In practice, we observe that MLLMs follow the formatting instructions well.

---

**Illustrations of Inputs and Outputs**

- {image_data}, {text_data}, and {question}: Images, texts, and questions from the datasets.
- {label_list_with_index}: A label list with indices of each label, for example, "0: airplane, 1: automobile, 2: bird, 3: cat, 4: deer, 5: dog, 6: frog, 7: horse, 8: ship, 9: truck".
- {first_label}: The first label in the label list, for example, "airplane".
- {label_index} & [label_index]: The label index generated by the annotator.
- [error_probability] & [error_level]: The error probability or level generated by the criticizer.
- {CoT_A} & [CoT_A]: The CoT generated by the annotator.
- [CoT]: The CoT generated by the criticizer.

---

**[Image Classification] Annotation Prompt - naïve**

**Prompt**
{image_data}
Determine the label of the image classification task. The list of labels is: {label_list_with_index}. The required output format is: [label_index]. For example, if the label is {first_label}, you should output [0]. Do not return other texts.
**Output**
[label_index]

---

**[Text Classification] Annotation Prompt - naïve**

**Prompt**
Determine the label of the text classification task. The text is {text_data}. The list of labels is: {label_list_with_index}. The required output format is: [label_index]. For example, if the label is {first_label}, you should output [0]. Do not return other texts.
**Output**
[label_index]

---

**[VQA-Binary] Annotation Prompt - naïve**

**Prompt**
{image_data}
Answer the question based on the given image. The question is: {question}. The required output format is [0] for No and [1] for Yes. Do not return other text.
**Output**
[answer_index]

---

**[Image Classification] Annotation Prompt - CoT**

**Prompt**
{image_data}
Determine the label of the image classification task. The list of labels is: {label_list_with_index}. The required output format is: [label_index]. Think step-by-step and provide your reasoning. Example of required output format is: [reasoning][label_index]. The first brackets contain your step-by-step reasoning, and the second brackets contain the label index such as [0] for {first_label}. Do not return other texts.
**Output**
[CoT_A][label_index]

**[Image Classification] Black-box Criticism Prompt - naïve**

**Prompt**

{image_data}

Your task is to produce the probability that the label of the given image is wrong. The list of labels is: {label_list_with_index}. The label of the image is: {label_index}. The required output format is: [error_probability]. For example, [0.911]. The error probability should be reported in 3 decimals. Do not return other texts.

**Output**

[error_probability]

---

**[Image Classification] Black-box Criticism Prompt - CoT**

**Prompt**

{image_data}

Your task is to produce the probability that the label of the given image is wrong. The list of labels is: {label_list_with_index}. The label of the image is: {label_index}. Think step-by-step and provide your reasoning. The required output format is: [reasoning][error_probability]. The first brackets contain your step-by-step reasoning, and the second brackets contain the error probability such as [0.911]. The error probability should be reported in 3 decimals. Do not return other texts.

**Output**

[CoT][error_probability]

---

**[Image Classification] Black-box Criticism Prompt - multiple choice**

**Prompt**

{image_data}

Your task is to analyze if label of the given image is wrong and select from [1: correct, 2: correct but not sure, 3: not sure, 4: incorrect but not sure, 5: incorrect]. The list of labels is: {label_list_with_index}. The label of the image is: {label_index}. Think step-by-step and provide your reasoning. The required output format is: [reasoning][error_level]. The first brackets contain your step-by-step reasoning, and the second brackets contain the error level such as [5] for incorrect. Do not return other texts.

**Output**

[CoT][error_level]

---

**[Image Classification] Black-box Criticism Prompt - devil's advocate**

**Prompt**

{image_data}

Your task is to produce the probability that the statement related to the label of the given image is wrong. The list of labels is: {label_list_with_index}. The statement of the image label is: [CoT_A]. Think step-by-step and provide your reasoning. The required output format is: [reasoning][error_probability]. The first brackets contain your step-by-step reasoning, and the second brackets contain the error probability such as [0.911]. The error probability should be reported in 3 decimals. Do not return other texts.

**Output**

[CoT][error_probability]

**[Image Classification] White-box Criticism Prompt - naïve**

**Prompt**

{image_data}

Your task is to decide whether the label of the given image is wrong. The list of labels is: {label_list_with_index}. The label of the image is: {label_index}. The required output is either Yes or No, where Yes means mistake and No otherwise. Do not return other texts.

**Output**

Yes/No

---

**[Image Classification] White-box Criticism Prompt - CoT**

**Prompt**

{image_data}

Your task is to decide whether the label of the given image is wrong. The list of labels is: {label_list_with_index}. The label of the image is: {label_index}. Think step-by-step and provide your reasoning. The required output format is: [reasoning][answer]. The first brackets contain your step-by-step reasoning, and the second brackets contain either Yes or No with Yes meaning mistake and No otherwise. Do not return other texts.

**Output**

[CoT][Yes/No]

# D  Theoretical Proofs

## D.1  Proof of Proposition 5.1 - Statistical Properties of ACT Loss (Variance)

*Proof.* Recall that the ACT loss is defined as:

$$\mathcal{L}_\theta^{(\text{ACT})} = \frac{1}{N} \sum_{i=1}^{N} \left( \ell_{\theta,i}^{(m)} + \left( \ell_{\theta,i} - \ell_{\theta,i}^{(m)} \right) \frac{\delta_i(B)}{\pi_B(\hat{\epsilon}_i)} \right), \tag{11}$$

where $\delta_i(B) \sim \mathbb{B}(\pi_B(\hat{\epsilon}_i))$. Then, we define

$$Z_i := \ell_{\theta,i}^{(m)} + \left( \ell_{\theta,i} - \ell_{\theta,i}^{(m)} \right) \cdot \frac{\delta_i(B)}{\pi_B(\hat{\epsilon}_i)}, \quad \text{and} \quad Z := \frac{1}{N} \sum_{i=1}^{N} Z_i = \mathcal{L}_\theta^{(\text{ACT})}.$$

Assuming that data are i.i.d, the variance of the ACT loss is now equivalent to the variance of $Z$, which is

$$\text{Var}(Z) = \text{Var}\left( \frac{1}{N} \sum_{i=1}^{N} Z_i \right) = \frac{1}{N^2} \sum_{i=1}^{N} \text{Var}(Z_i). \tag{12}$$

Hence, it suffices to only calculate the variance of $Z_i$, which can be decomposed into two parts as $\text{Var}(Z_i) = \mathbb{E}[Z_i^2] - (\mathbb{E}[Z_i])^2$. We first expand:

$$
\begin{aligned}
Z_i^2 &= \left( \ell_{\theta,i}^{(m)} + \left( \ell_{\theta,i} - \ell_{\theta,i}^{(m)} \right) \cdot \frac{\delta_i(B)}{\pi_B(\hat{\epsilon}_i)} \right)^2 \\
&= \left( \ell_{\theta,i}^{(m)} \right)^2 + 2\ell_{\theta,i}^{(m)}(\ell_{\theta,i} - \ell_{\theta,i}^{(m)}) \cdot \frac{\delta_i(B)}{\pi_B(\hat{\epsilon}_i)} + (\ell_{\theta,i} - \ell_{\theta,i}^{(m)})^2 \cdot \frac{\delta_i(B)}{\pi_B(\hat{\epsilon}_i)^2},
\end{aligned}
\tag{13}
$$

where we have $\delta_i^2(B) = \delta_i(B)$ because $\delta_i(B)$ is either 0 or 1. We assume that $\delta_i(B)$ and $\ell_{\theta,i}$ are independent. In addition, it is easy to see that $\mathbb{E}[\delta_i(B)] = \pi_B(\hat{\epsilon}_i)$, and that $\mathbb{E}[Z_i] = \mathbb{E}[\ell_{\theta,i}]$. So, we calculate the expectation of $Z_i^2$ as follows:

$$
\begin{aligned}
\mathbb{E}[Z_i^2] &= \left( \ell_{\theta,i}^{(m)} \right)^2 + 2\ell_{\theta,i}^{(m)} \mathbb{E}\left[ \ell_{\theta,i} - \ell_{\theta,i}^{(m)} \right] + \mathbb{E}\left[ \left( \ell_{\theta,i} - \ell_{\theta,i}^{(m)} \right)^2 \cdot \frac{1}{\pi_B(\hat{\epsilon}_i)} \right] \\
&= \mathbb{E}\left[ \ell_{\theta,i}^2 - \left( \ell_{\theta,i} - \ell_{\theta,i}^{(m)} \right)^2 \right] + \mathbb{E}\left[ \left( \ell_{\theta,i} - \ell_{\theta,i}^{(m)} \right)^2 \cdot \frac{1}{\pi_B(\hat{\epsilon}_i)} \right] \\
&= \mathbb{E}\left[ \ell_{\theta,i}^2 \right] + \mathbb{E}\left[ \left( \ell_{\theta,i} - \ell_{\theta,i}^{(m)} \right)^2 \cdot \left( \frac{1}{\pi_B(\hat{\epsilon}_i)} - 1 \right) \right]
\end{aligned}
\tag{14}
$$

Thus, we have

$$
\begin{aligned}
\text{Var}(Z_i) &= \mathbb{E}\left[ \ell_{\theta,i}^2 \right] + \mathbb{E}\left[ \left( \ell_{\theta,i} - \ell_{\theta,i}^{(m)} \right)^2 \cdot \left( \frac{1}{\pi_B(\hat{\epsilon}_i)} - 1 \right) \right] - \mathbb{E}^2[\ell_{\theta,i}] \\
&= \text{Var}(\ell_{\theta,i}) + \mathbb{E}\left[ \left( \ell_{\theta,i} - \ell_{\theta,i}^{(m)} \right)^2 \cdot \left( \frac{1}{\pi_B(\hat{\epsilon}_i)} - 1 \right) \right]
\end{aligned}
\tag{15}
$$

Finally, we can show that

$$
\begin{aligned}
\text{Var}\left( \mathcal{L}^{(ACT)} \right) &= \text{Var}(Z) = \frac{1}{N^2} \sum_{i=1}^{N} \text{Var}(Z_i) \\
&= \frac{1}{N^2} \sum_{i=1}^{N} \left( \text{Var}(\ell_{\theta,i}) + \mathbb{E}\left[ \left( \ell_{\theta,i} - \ell_{\theta,i}^{(m)} \right)^2 \cdot \left( \frac{1}{\pi_B(\hat{\epsilon}_i)} - 1 \right) \right] \right) \\
&= \frac{1}{N} \left( \text{Var}(\ell_\theta) + \mathbb{E}\left[ \left( \ell_\theta - \ell_\theta^{(m)} \right)^2 \cdot \left( \frac{1}{\pi_B(\hat{\epsilon})} - 1 \right) \right] \right).
\end{aligned}
\tag{16}
$$

This completes the proof. $\square$

## D.2 Proof of Theorem 5.2 - Probabilistic Upper Bound of the Parameter Gap

*Proof.* From the definition of ACT loss in Equation (11), we can show that

$$
\nabla \mathcal{L}_\theta^{(ACT)} = \frac{1}{N} \sum_{i=1}^N \pi_i \nabla \ell_{\theta,i} + \frac{1}{N} \sum_{i=1}^N (1 - \pi_i) \nabla \ell_{\theta,i}^{(m)}
$$

$$
= \nabla \mathcal{L}_\theta + \left( \frac{1}{N} \sum_{i=1}^N \pi_i \nabla \ell_{\theta,i} - \frac{1}{N} \sum_{i=1}^N \nabla \ell_{\theta,i} \right) + \frac{1}{N} \sum_{i=1}^N (1 - \pi_i) \nabla \ell_{\theta,i}^{(m)}
$$

$$
= \nabla \mathcal{L}_\theta + \frac{1}{N} \sum_{i=1}^N (1 - \pi_i) \left( \nabla \ell_{\theta,i}^{(m)} - \nabla \ell_{\theta,i} \right) \tag{17}
$$

where we let $\pi_i = \frac{\delta_i(B)}{\pi_B(\hat{\epsilon}_i)}$ and $\mathcal{L}_\theta = \frac{1}{N} \ell_{\theta,i}$.

It is easy to see $\mathbb{E}\left[ (1 - \pi_i) \left( \nabla \ell_{\theta,i}^{(m)} - \nabla \ell_{\theta,i} \right) \right] = 0$. Assume there exist constants $q, C > 0$ such that the transformed error $\pi_B(\hat{\epsilon}_i) \geq q$ for all $i$ with $\delta_i(B) = 1$, and the gradient gap $\left\| \nabla \ell_{\theta,i}^{(m)} - \nabla \ell_{\theta,i} \right\| \leq C$ for all $i \in \{1, 2, ..., N\}$. Next, we bound for any $i$ that

$$
\left\| (1 - \pi_i) \left( \nabla \ell_{\theta,i}^{(m)} - \nabla \ell_{\theta,i} \right) \right\| \leq |1 - \pi_i| \left\| \nabla \ell_{\theta,i}^{(m)} - \nabla \ell_{\theta,i} \right\| \leq \max \left\{ 1, \frac{1-q}{q} \right\} C =: c_0,
$$

$$
\text{and} \quad \mathbb{E} \left\| (1 - \pi_i) \left( \nabla \ell_{\theta,i}^{(m)} - \nabla \ell_{\theta,i} \right) \right\|^2 \leq \mathbb{E}\left[ (1 - \pi_i)^2 \right] \left\| \nabla \ell_{\theta,i}^{(m)} - \nabla \ell_{\theta,i} \right\|^2 \leq \frac{1-q}{q} C^2 =: c_1.
$$

Then we apply the vector Bernstein's inequality (e.g., Lemma 18 in [59]) such that

$$
\mathbb{P}\left( \left\| \frac{1}{N} \sum_{i=1}^N (1 - \pi_i) \left( \nabla \ell_{\theta,i}^{(m)} - \nabla \ell_{\theta,i} \right) \right\| \geq \epsilon \right) \leq 2 \exp\left( -N\epsilon^2 / (8c_1) \right) \tag{18}
$$

for $0 < \epsilon < \frac{c_1}{c_0}$. Then, with a probability of at least $1 - p$ where $p \in (0, 1)$, we have

$$
\left\| \frac{1}{N} \sum_{i=1}^N (1 - \pi_i) \left( \nabla \ell_{\theta,i}^{(m)} - \nabla \ell_{\theta,i} \right) \right\| \leq \sqrt{\frac{8c_1 \log(2/p)}{N}} \tag{19}
$$

for any $N \geq 8c_0^2 \log(2/p)/c_1$.

Finally, due to the $\mu$-strong convexity of $\ell_{\theta,i}^{(\cdot)}$ and thus $\mathcal{L}_\theta^{(\cdot)}$, with a probability of at least $1 - p$, we can bound the parameter gap

$$
\left\| \theta_*^{(ACT)} - \theta_* \right\| \leq \frac{1}{\mu} \left\| \nabla \mathcal{L}_{\theta_*^{(ACT)}}^{(ACT)} - \nabla \mathcal{L}_{\theta_*}^{(ACT)} \right\|
$$

$$
= \frac{1}{\mu} \left\| \nabla \mathcal{L}_{\theta_*} - \nabla \mathcal{L}_{\theta_*}^{(ACT)} \right\|
$$

$$
= \frac{1}{\mu} \left\| \frac{1}{N} \sum_{i=1}^N (1 - \pi_i) \left( \nabla \ell_{\theta_*,i}^{(m)} - \nabla \ell_{\theta_*,i} \right) \right\|
$$

$$
\leq \sqrt{\frac{8c_1 \log(2/p)}{\mu^2 N}} \tag{20}
$$

where $\theta_*^{(ACT)} = \arg\min_\theta \mathcal{L}_\theta^{(ACT)}$, and $\theta_* = \arg\min_\theta \mathcal{L}_\theta$.

This completes the proof. $\qquad \square$

# E   Further Details of ACT Losses

## E.1   ACT Losses with Different Sampling Rules

The ACT losses with different sampling rules are listed as follows:

- *Normalization* [17, 20]

$$\mathcal{L}_\theta^{(\text{ACT})} = \frac{1}{N} \sum_{i=1}^{N} \left( \lambda \ell_{\theta,i}^{(m)} + \left( \ell_{\theta,i} - \lambda \ell_{\theta,i}^{(m)} \right) \times \frac{\delta_i(B) \times \sum_{n=1}^{N} \hat{\epsilon}_n}{B \times \hat{\epsilon}_i} \right); \qquad (21)$$

- *Exponential Weighting*

$$\mathcal{L}_\theta^{(\text{ACT})} = \frac{1}{N} \sum_{i=1}^{N} \left( \lambda \ell_{\theta,i}^{(m)} + \left( \ell_{\theta,i} - \lambda \ell_{\theta,i}^{(m)} \right) \times \delta_i(B) \left( 1 + e^{-\beta(\hat{\epsilon}_i - \alpha)} \right) \right); \qquad (22)$$

- *Thresholding*

$$\mathcal{L}_\theta^{(\text{ACT})} = \frac{1}{N} \sum_{i=1}^{N} \left( \lambda \ell_{\theta,i}^{(m)} + \left( \ell_{\theta,i} - \lambda \ell_{\theta,i}^{(m)} \right) \times \delta_i(B) \right), \qquad (23)$$

where $\lambda \in [0,1]$ is the power tuning parameter [17, 60], which controls the extent to which machine annotations are utilized. Specifically, $\lambda = 0$ corresponds to completely ignoring machine annotations, while $\lambda = 1$ corresponds to the full usage. Notably, when employing the thresholding sampling rule with $\lambda = 1$, the ACT loss is equivalent to the standard Cross-entropy loss computed using human-annotated labels when available, and machine-annotated labels otherwise. Therefore, we stated in Section 5.2 that the Cross-entropy loss is a special case of the ACT loss.

## E.2   Distributions of Transformed Errors with Different Sampling Rules

In Figure 10, we present the distributions of transformed errors for data samples reviewed by humans ($\delta(B) = 1$) under different sampling strategies. We observe that, with normalization sampling, the lower bounds of the transformed errors are close to 0 across all presented datasets. In contrast, for exponential weighting, the lower bounds typically around 0.8, while thresholding yields a consistent lower bound of 1.0. Based on Theorem 5.2, these results provide an explanation for why exponential weighting and thresholding can lead to better downstream training performance.

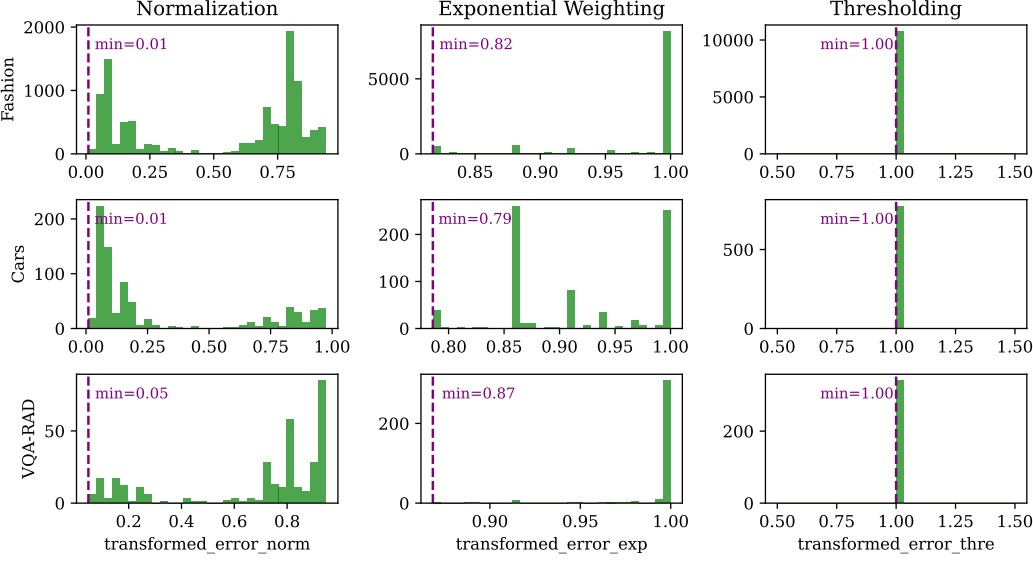

Figure 10: Distributions of transformed errors with different sampling rules ($\delta(B) = 1$). $B$ is set to the ideal human budget for each dataset.

# F    Sensitivity Analyses of Human Budget

## F.1    AQG vs. Human Budget

In Figure 11, we illustrate how Annotation Quality Gain (AQG) changes with the human budget proportion. We observe that as the human budget increases, AQG generally rises rapidly at first, then begins to plateau after a certain point. This initial rapid increase suggests that the human-corrected samples tend to be more obvious and easily identifiable errors. On most datasets, AQG does not reach 100% before the human budget reaches its maximum. In other words, it is usually difficult to achieve perfect annotation quality without reviewing all examples. This indicates that some subtle or hard-to-detect errors are unavoidable. However, we will show in Figure 12 that this does not undermine the effectiveness of using ACT to reduce human effort. With the ACT loss, a promising downstream training does not rely on perfectly labeled data.

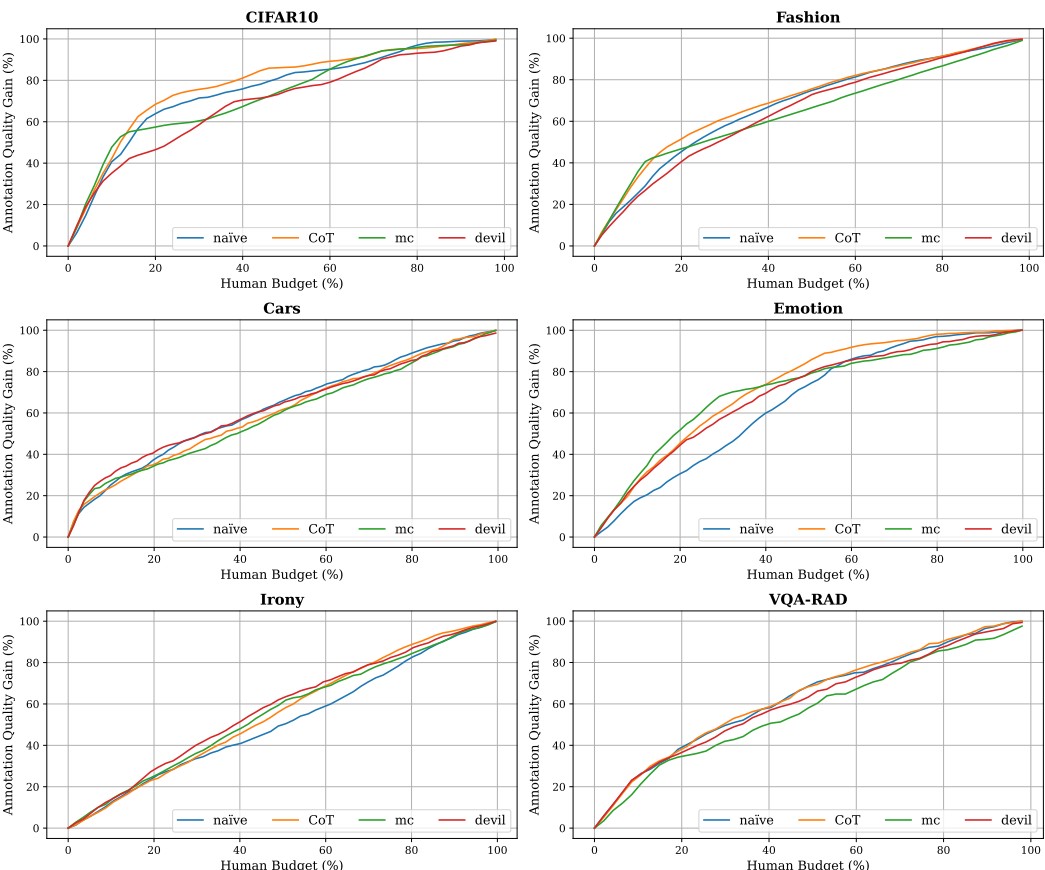

Figure 11: How AQG changes with human budget (%). The results are presented for GPT4o self-criticism with 4 black-box strategies.

## F.2 Downstream Performance vs. Human Budget

We conducted the human budget sensitivity experiments using GPT4o self-criticism and the thresholding sampling rule. The results are shown in Figure 12. We observe that both annotation accuracy and downstream accuracy generally increase with a larger human budget. When using the ideal budget, a performance gap can be observed across all datasets. This is because the criticizer is not perfectly accurate, leading to some overlooked mislabeled data, which slightly degrades the final training performance. In Figure 12, we also show the downstream performance gain achieved by adding a 10% buffer budget on top of the ideal budget. In 4 out of 6 datasets, this buffer nearly eliminates the performance gap, while the gap is significantly reduced in the other 2 datasets. **Therefore, we recommend first evaluating the annotator's accuracy, and then adding a reasonable buffer to the ideal budget based on the observed error rate.**

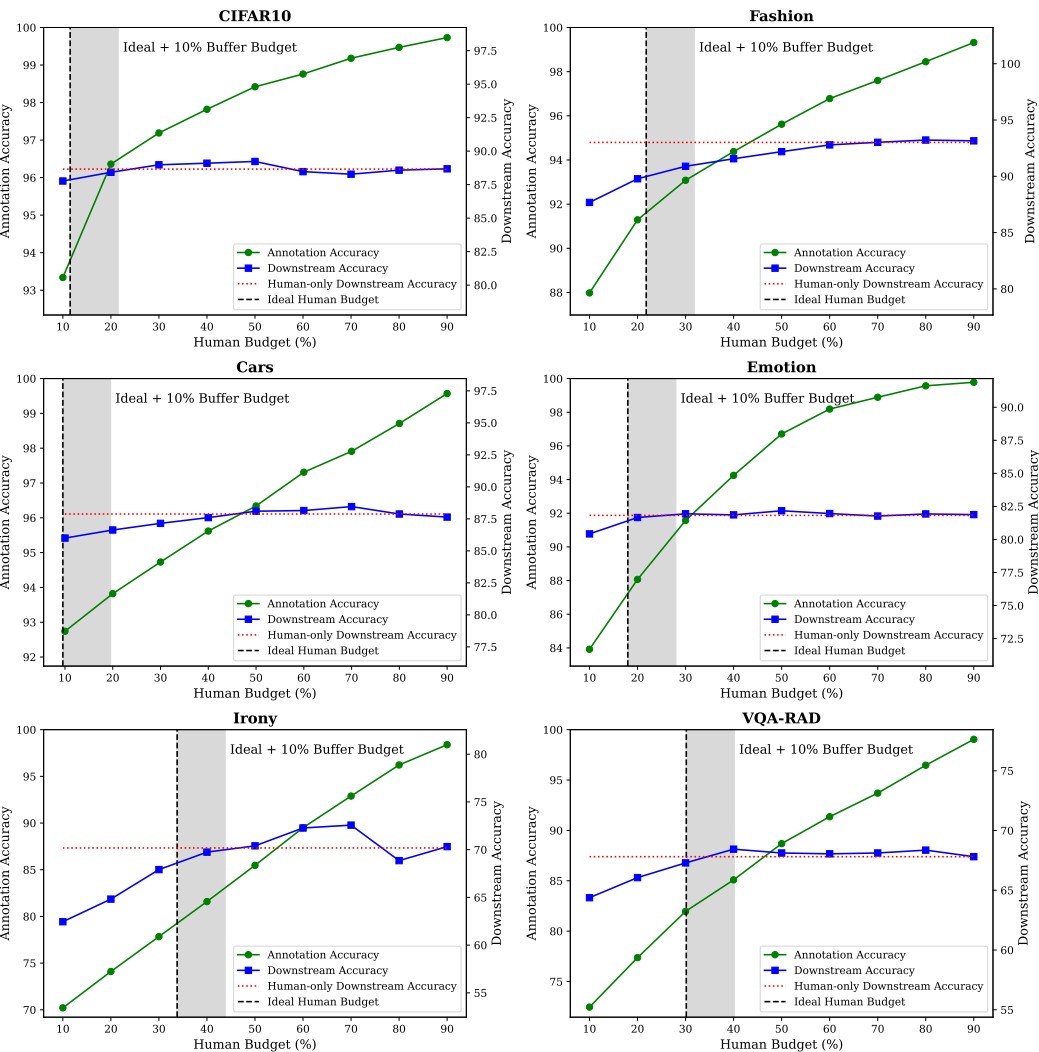

Figure 12: How annotation accuracy and downstream accuracy change with the human budget (%). The black vertical line shows the position of the ideal human budget (i.e., one minus the initial annotator accuracy). The grey area shows the buffer budget over the ideal budget.

# G Extended Discussions and Experiments

## G.1 Additional Related Works

**Enhance Data Annotation with LLMs**

To improve the quality of LLM annotation, AdaICL adopts in-context learning (ICL) with examples annotated by human [18]. The ICL examples are actively selected based on LLM logit probabilities during annotation, which means their method only supports white-box LLMs. In addition, AdaICL lacks mechanisms for long-context visual inference, making it difficult to apply directly to vision-based tasks. This is because retrieving and encoding a large number of visual examples during inference would incur prohibitive computational costs [61, 62]. However, we consider ICL as a potential direction for future work, particularly as a component in our data pipeline for building an annotator.

Other related works typically follow a three-step LLM-human collaborative framework: (1) LLMs generate initial labels, (2) a verifier assesses the correctness of these labels and outputs verification scores, and (3) human annotators re-annotate a selected subset of labels based on these scores [15, 16, 17, 19]. The primary distinction among these methods lies in the design of the verifier. For instance, Model-in-the-Loop (MILO) [19] utilizes the logit scores from another LLM-based verifier (similar to our white-box criticizer). In contrast, MEGAnno+ [15] directly employs the logit probabilities from the LLM annotator itself. Another framework proposed by [19] uses a verifier implemented as a Support Vector Machine [63], Random Forests [64], or BERT [65], trained on additional human-annotated data. Unlike our approach, the aforementioned methods focus solely on annotation accuracy without considering the utility of annotations for downstream training. This narrow focus limits their effectiveness, as high-accuracy labels do not necessarily translate into meaningful model improvements.

The most relevant work to our research is CDI [17], which identifies LLM errors using a trained XGBoost model [49] and relies on human annotators for correction. During the annotation process, CDI prompts annotators to provide both labels and corresponding verbalized confidence scores. These confidence scores are provided in a black-box manner, where higher values indicate greater confidence. For example, an annotator might respond, "The label is cat, and my confidence is 0.999," to express high certainty. The XGBoost model then uses these confidence scores as input to learn and predict error probabilities. The ground truth is either 0 or 1 depending on the correctness of the annotation, and then the logit probabilities are regarded as error probabilities. However, CDI has two key limitations: (1) its error detection mechanism lacks flexibility, requiring task-specific design and additional training data, and (2) it employs a normalization-based active M-estimation loss, which we find suboptimal in downstream tasks.

## G.2 Potential Improvements of ACT

Here, we outline several potential improvements to the ACT data pipeline, as inspired by the related works. First, drawing from AdaICL [18], we could enhance the prompts of the MLLM annotator—particularly for NLP tasks—by incorporating in-context examples. Second, following the approach of MEGAanno+ [15], it may be beneficial to combine the annotator's confidence scores with the criticizer's error estimations to better capture the insights from both perspectives. Finally, while the current pipeline relies on a single MLLM annotator–criticizer pair, it could be extended to a multi-model setup using techniques such as majority voting or peer discussion in [13]. An illustration is provided in Figure 13.

For decision of the human budget, one may consider dynamically estimating and adjusting the human budget during the review process. As more human-verified labels are accumulated over time, it becomes increasingly feasible to refine the error rate estimation and update the budget allocation accordingly.

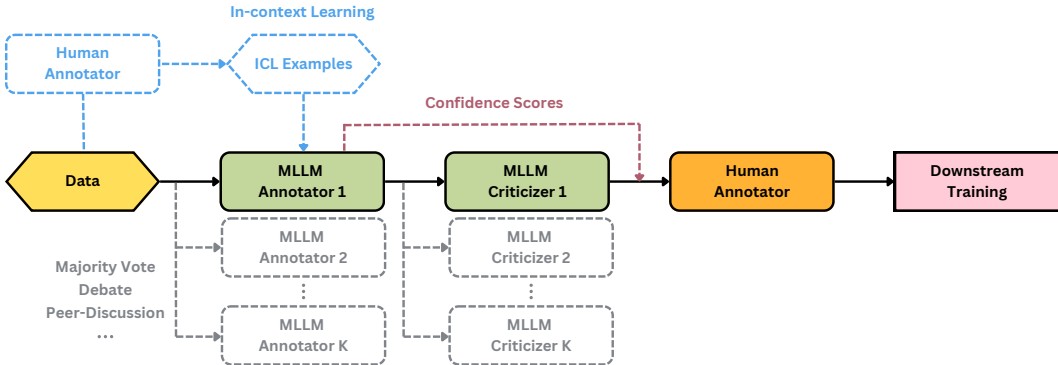

Figure 13: Illustration of potential improvements of the ACT data pipeline.

## G.3 Supplementary Experiments

### G.3.1 Comparison with CDI

In Table 6, we compare the downstream performance of models trained on ACT and CDI data using various loss functions. Note that CDI data with the ACT norm. loss is corresponding to the approach proposed in [17]. We apply the same human budget (the ideal human budget) to both ACT and CDI. For CDI, a proportion of this budget is allocated to training the XGBoost error detector. The results demonstrate that ACT consistently outperforms CDI in reducing the downstream performance gap.

Table 6: Comparison between ACT and CDI in downstream tasks. The test accuracy (%) is reported in form of mean ± std over 5 runs.

| Training Data - Loss | CIFAR10 (ResNet18) | Fashion (ResNet18) | Cars (ResNet18) | Emotion (RoBERTa) | Irony (RoBERTa) | VQA-RAD (BLIP-VQA) |
|---|---|---|---|---|---|---|
| Human only - Cross-entropy Loss | 88.66 ± 0.97 | 93.01 ± 0.63 | 87.88 ± 0.36 | 81.82 ± 0.57 | 70.18 ± 3.23 | 67.81 ± 1.47 |
| CDI data - Cross-entropy loss | 84.02 ± 0.85 | 86.99 ± 0.72 | 85.61 ± 0.24 | 79.91 ± 1.37 | 66.63 ± 3.44 | 61.77 ± 3.41 |
| CDI data - ACT norm. loss | 72.22 ± 1.71 | 83.38 ± 1.79 | 10.76 ± 1.12 | 79.05 ± 1.15 | 65.96 ± 3.36 | 62.24 ± 3.05 |
| CDI data - ACT exp. loss | 84.99 ± 0.31 | 87.72 ± 0.54 | 86.03 ± 0.15 | 80.51 ± 1.49 | 68.44 ± 2.22 | 67.33 ± 1.66 |
| CDI data - ACT thre. loss | 84.91 ± 0.57 | 87.64 ± 0.52 | 85.89 ± 0.27 | 80.00 ± 0.83 | 68.19 ± 2.29 | 67.44 ± 2.16 |
| ACT data - Cross-entropy loss | 85.59 ± 0.52 | 87.50 ± 0.86 | 85.88 ± 0.26 | 80.82 ± 1.08 | 67.83 ± 2.82 | 61.83 ± 3.27 |
| ACT data - ACT norm. loss | 64.70 ± 5.46 | 69.27 ± 7.25 | 11.54 ± 0.96 | 79.87 ± 0.88 | 65.66 ± 2.00 | 62.55 ± 3.01 |
| **ACT data - ACT exp. loss (Ours)** | 87.73 ± 0.36 | 89.73 ± 0.35 | 86.19 ± 0.14 | 81.44 ± 0.51 | 68.49 ± 3.20 | 67.73 ± 1.33 |
| **ACT data - ACT thre. loss (Ours)** | 87.95 ± 0.35 | 89.16 ± 0.89 | 86.00 ± 0.26 | 81.41 ± 0.64 | 68.21 ± 1.94 | 67.02 ± 1.32 |
| Human-CDI performance gap (%) | 3.67% | 5.29% | 1.85% | 1.31% | 1.74% | 0.37% |
| Human-ACT performance gap (%) | 0.71% | 3.28% | 1.69% | 0.38% | 1.69% | 0.08% |
| Human budget (%) | 11.52% | 21.81% | 9.56% | 17.98% | 33.79% | 30.15% |

### G.3.2 Comparison with Active Learning and Pseudo Labels

We compare our method with more existing baselines from related fields on CIFAR10 (ResNet18):

- **Pseudo Labelling**: We randomly label 11.52% of the data to fine-tune ResNet18, then iteratively add high-confidence predictions (e.g., logit probability > 0.9) as pseudo labels to the training set.

- **Active Pseudo Labelling**: We start by labelling 100 samples per class and progressively select uncertain samples (based on prediction entropy) for annotation until the human budget is exhausted.

The results shown in Table 7 indicate that our approach outperforms these two methods by 4.28% and 2.56% in terms of performance gap from 100% human annotation.

Table 7: Comparison with baselines from related fields.

| Method | Accuracy (%) | Human Performance Gap (%) |
|---|---|---|
| Human-only – CE loss | $88.66 \pm 0.97$ | – |
| ACT – ACT thre. loss (Ours) | $87.95 \pm 0.35$ | 0.71 |
| Pseudo labeling – CE loss | $83.67 \pm 0.32$ | 4.99 |
| Active Pseudo labeling – CE loss | $85.39 \pm 0.91$ | 3.27 |

### G.3.3 More Challenging Tasks: Biased and Long-tail Data

To validate the effectiveness of our method in more challenging tasks, we conduct two experiments:

- AlleNoise [66] is a product classification dataset where similar category names often result in subjective and biased labelling. We focus on 20 labels under the "Sports & Travel" category, which exhibit approximately 70% label similarity—making the task ambiguous and prone to bias. Additionally, the largest category contains 27 times more samples than the smallest, reflecting a significant long-tail distribution. The results are shown in Table 8, demonstrating that ACT yields only a 1.4% performance gap in accuracy and a 0.0041 difference in F1-score compared to full human annotation.

Table 8: Experiment results on AlleNoise with RoBERTa.

| Data – Loss (AlleNoise – RoBERTa) | Accuracy (%) | F1-score |
|---|---|---|
| Human annotation – CE loss | $91.54 \pm 1.84$ | $0.8448 \pm 0.0165$ |
| ACT data – ACT thre. loss (GPT4o self-criticism) | $90.14 \pm 1.79$ | $0.8397 \pm 0.0191$ |
| **Human–ACT performance gap** | 1.40 | 0.0041 |

- CIFAR10-LT [3] is a long-tailed variant of CIFAR10 with an imbalance ratio of 100 between the most and least frequent classes. As shown in Table 9, ACT achieves a performance gap of 1.97% in accuracy and 0.0148 in F1-score compared to full human annotation.

Table 9: Experiment results on CIFAR10-LT with ResNet.

| Data – Loss (CIFAR10-LT – ResNet) | Accuracy (%) | F1-score |
|---|---|---|
| Human annotation – CE loss | $70.29 \pm 2.15$ | $0.6820 \pm 0.0261$ |
| ACT data – ACT thre. loss (GPT4o self-criticism) | $68.32 \pm 1.66$ | $0.6672 \pm 0.0263$ |
| **ACT – Human performance gap** | 1.97 | 0.0148 |

---

[3] https://huggingface.co/datasets/tomas-gajarsky/cifar10-lt

### G.3.4 Analyses on Critic True Positives and False Positives

### a. Analyze Criticism Reliability with False Positives

Table 10 reports the number of false positives produced by different error detection methods across six datasets. The human budget is consistent with Table 3. We compare GPT4o self-criticism and the best-performing cross-criticizer for each task, with random error sampling included as a baseline. We observe the following:

- MLLM-based error detection significantly outperforms random sampling, consistently yielding fewer false positives across tasks.

- False positives account for approximately 5%–15% of the total dataset, depending on the complexity and ambiguity of the task.

These results suggest that MLLM-based criticism is an effective strategy for identifying annotation errors. In addition, there remains room for improvement, especially through incorporating sample-specific analysis and adaptive selection mechanisms in real-world applications.

Table 10: Number of false positives by different error detection methods across six datasets.

| Error Detection Methods | CIFAR10 (data size~50k) | Fashion (data size~60k) | Cars (data size~8k) |
|---|---|---|---|
| Random Sampling | 4866 | 9852 | 704 |
| GPT4o Self-Criticizer | 2805 | 6395 | 594 |
| Optimal Cross-Criticizer | 2692 | 6087 | 514 |

| Error Detection Methods | Emotion (data size~3k) | Irony (data size~3k) | VQA-RAD (data size~1k) |
|---|---|---|---|
| Random Sampling | 480 | 640 | 198 |
| GPT4o Self-Criticizer | 302 | 591 | 123 |
| Optimal Cross-Criticizer | 299 | 455 | 113 |

### b. Compare Self- and Cross-Criticism with True Positives

We use dataset VQA-RAD as an example, where GPT4o only achieves 69.85% annotation accuracy. The human budget is set to 30.15%. In Table 11, we present the number of true positives and annotation accuracy after human correction. Randomly sampled errors are included as a baseline. It is clear that all MLLM criticizers perform much better than random sampling, while self- and cross-criticism do not show huge differences. This further strengthens the claim that self-criticism performs competitively with cross-criticism.

Table 11: Number of true positives and annotation accuracy after human correction (VQA-RAD).

| Error Detection Methods | #True Positives | Final Annotation Accuracy |
|---|---|---|
| Random Sampling | 85 | 78.89% |
| GPT4o (Self-criticism) | 160 | 86.87% |
| Gemini1.5P (Cross-criticism) | 153 | 86.12% |
| Claude3.5S (Cross-criticism) | 159 | 86.76% |
| LLaVA-OV (Cross-criticism) | 158 | 86.66% |
| Qwen2.5VL (Cross-criticism) | 170 | 87.94% |
| InternVL2.5 (Cross-criticism) | 149 | 85.70% |

### G.3.5 Ablation of Annotator-Criticizer Selection

We conduct ablation experiments that examine how annotator-criticizer choice affects final downstream model performance. The experiments are conducted with Cars, because variations in self- and cross-criticism performance are large on this dataset based on Figure 3. So, we would like to know whether the downstream performance also varies a lot.

In Table 12, we find that the effectiveness of criticism is indeed related to downstream performance. The best-performing criticizer (highest ABS) yields the smallest performance gap. However, the default GPT4o also performs reasonably well, keeping the gap within 2%.

Table 12: Ablation on how annotator-criticizer choice affects final downstream model performance. The results are reported in ABS (%) and accuracy (%).

| Cars – GPT4o Annotation | GPT4o | Gemini1.5P | Claude3.5S | LLaVA-OV | Qwen2.5VL | InternVL2.5 |
|---|---|---|---|---|---|---|
| Criticism ability – ABS | 60.6 | **70.9** | 53.2 | 55.9 | 54.4 | 57.5 |
| ACT data – ACT thre. loss | $86.00 \pm 0.26$ | $\mathbf{86.21 \pm 0.31}$ | $85.21 \pm 0.35$ | $85.39 \pm 0.29$ | $85.80 \pm 0.29$ | $85.45 \pm 0.31$ |
| Human–ACT performance gap | 1.88 | **1.67** | 2.67 | 2.49 | 2.08 | 2.43 |

### G.3.6 Cost-Benefit Analysis

We present a cost-benefit analysis using CIFAR10 (50k samples) to compare three annotation strategies: (1) 100% human annotation, (2) ACT with GPT4o self-criticism, and (3) ACT with GPT4o & Qwen2.5VL cross-criticism. Note that we exclude the cases where GPUs are not rented but owned, whose costs are difficult to estimate.

As shown in Table 13, API-based approaches are the most cost-efficient. We also highlight that monetary costs are not the only concern. Human annotation also incurs significant time and educational costs, further highlighting the efficiency of our machine-based approach.

Table 13: Cost comparison between human-only annotation and ACT approaches.

| Item | Human-only | ACT (GPT4o-GPT4o) | ACT (GPT4o-Qwen2.5VL) |
|---|---|---|---|
| **GPT4o Annotation** API costs (OpenAI) | – | $\sim$200 tokens $\times$ 10 / 1M = 0.002/image. Total = 50,000 $\times$ 0.002 = **$100** | Same as left $\rightarrow$ **$100** |
| **GPT4o CoT Criticism** API costs (OpenAI) | – | $\sim$200 tokens $\times$ 10 / 1M = 0.002/image. Total = 50,000 $\times$ 0.002 = **$100** | – |
| **Qwen2.5VL CoT Criticism** GPU rent costs (RunPod) | – | – | 8 A100 GPUs $\times$ 144 hrs $\times$ $1.74/hr $\approx$ **$2000** |
| **Human Annotation / Correction** Human costs (AWS avg.) | 50,000 images $\times$ $0.08 = **$4000** | 10% $\times$ 50,000 = 5,000 images $\rightarrow$ 5,000 $\times$ $0.08 = **$400** | Same as left $\rightarrow$ **$400** |
| **Total Costs** | **$4000** | **$600** | **$2500** |

### G.4    Extend ACT to More Complex Tasks

For tasks where the notion of "correct" and "incorrect" becomes subjective, the criticizer can output a continuous quality score or a relative ranking rather than a binary error probability.

The criticizer can leverage a strong evaluator, e.g., an MLLM-based evaluator or a reward model, to detect semantic inconsistencies, factual mistakes, or language issues in each candidate output and return a numerical score or rank.

For content-dense tasks such as multi-fact text summarisation, we can first segment the summary into individual factual statements and then evaluate each statement separately. This allows human to focus only on the segments flagged as uncertain, rather than re-checking the entire summary.

### G.5    Ethical Considerations

A potential ethical consideration of ACT lies in its influence on traditional annotation jobs, as the reduction of human effort could be perceived as a threat to existing human roles. However, the primary motivation of ACT is to alleviate the scalability limitations inherent in fully manual annotation, rather than to replace human annotators. By design, ACT facilitates human–AI collaboration, assigning human oversight to high-risk or ambiguous cases where nuanced judgment is essential. Importantly, this approach may also help mitigate the psychological burden associated with labeling harmful or sensitive content (e.g., depictions of violence or nudity), since human intervention is required only for a limited subset of critical instances.

