# OpenReview forum: "ACT as Human: Multimodal Large Language Model Data Annotation with Critical Thinking"
_NeurIPS.cc/2025/Conference — NeurIPS 2025 poster_

### Official Review · Reviewer_CvVf · 2025-07-02

**Clarity:** 2
**Significance:** 3
**Originality:** 3
**Rating:** 4
**Confidence:** 3

**Summary:**

This paper proposes the **Annotation with Critical Thinking (ACT)** data annotation pipeline to address the high cost and inefficiency of manual data labeling. The authors utilize multimodal large language models (MLLMs) as both annotators and critics. By leveraging a criticism mechanism to identify potential errors, human efforts are focused on reviewing only the most "suspicious" samples, improving annotation efficiency.

**Questions:**

1. **Model Selection and Generalization**:
   - Does ACT’s performance depend on specific MLLM models for different tasks? Could smaller-scale models achieve similar results?
   - Has the ACT pipeline been tested on open-domain datasets or real-world long-tail tasks for better generalization?
2. **Criticism Mechanism Improvements**:
   - What are the performance differences between self-criticism and cross-criticism for more complex tasks? Are there potential ways to enhance the criticism mechanism further?
3. **Trade-offs in Annotation Quality**:
   - Since CoT (Chain-of-Thought) improves criticism but not annotation, could optimized CoT strategies enhance annotation quality?
4. **Sampling Strategy Selection**:
   - How should sampling strategies (e.g., thresholding or exponential weighting) be chosen in practice? Are there automated methods for parameter selection?
5. **Social and Ethical Considerations**:
   - What are the potential impacts of scaling up the ACT pipeline on human annotators? Have ethical concerns related to such impacts been considered?

### **Suggestions for Improvement**:
- Conduct experiments on more complex tasks and data distributions to better evaluate the ACT pipeline’s applicability.
- Provide a more comprehensive discussion of societal and ethical impacts, such as the potential effects on the annotation industry.

**Ethical Concerns:**

["NO or VERY MINOR ethics concerns only"]

**Final Justification:**

The responses have clarified several key concerns I had about your work. The additional experiments on long-tail datasets, clarification about multiple independent runs, expanded ethical considerations, and comparison between self-criticism and cross-criticism mechanisms help strengthen your paper. While there remain opportunities to further explore the stability of criticism mechanisms across more diverse tasks and model scales, your current results showing reasonable performance gaps compared to human annotation while significantly reducing annotation effort are promising.

**Limitations:**

yes

**Paper Formatting Concerns:**

No major formatting issues found.

**Quality:**

2

**Strengths And Weaknesses:**

### **Strengths**
1. **Theoretical and Practical Contributions**:
   - Provides a strong theoretical framework for improving annotation quality through criticism mechanisms.
   - Empirical results demonstrate that models trained on ACT data achieve performance gaps of less than 2% compared to fully human-annotated datasets.
2. **Resource Efficiency**:
   - Focuses human review efforts on high-risk samples, reducing total human costs by 70%-90%.
   - Sampling strategies (e.g., thresholding, exponential weighting) effectively improve resource utilization.
3. **Practical Usability**:
   - Offers detailed user guidelines and actionable recommendations for real-world deployment.

### **Weaknesses**
1. **Criticism Mechanism Limitations**:
   - While the paper explores self-criticism and cross-criticism, it lacks in-depth discussion on their stability across tasks and model scales.
2. **Limited Experiment Scope**:
   - Experiments focus on standard datasets (e.g., CIFAR-10, Fashion-MNIST), leaving the applicability to more complex or long-tail datasets unverified.
   - Benchmark experiments are based on single runs, which, despite stability analysis, could benefit from multiple independent runs for stronger statistical validity.
3. **Theoretical Assumption Issues**:
   - The method assumes human-annotated labels closely approximate ground truth, which may not hold in noisy or biased scenarios.
4. **Lack of Societal Impact Discussion**:
   - The paper lacks a thorough discussion of potential negative societal impacts, such as effects on human annotators or privacy concerns.

---

> ### Author Rebuttal · Authors · 2025-07-30
>
> Dear Reviewer CvVf,
>
> We sincerely thank you for the thoughtful feedback and for recognizing both the theoretical and practical significance of our method. We are especially grateful for the mention of ethical issues. This gives us a valuable opportunity to discuss the ethical implications of our approach in real-world applications. In addition, we appreciate the suggestion to consider long-tail data, which helps us to enrich the analyses of our method. Please see below for the modifications we have made in response to your feedback, as well as clarifications to address potential misunderstandings.
>
> &nbsp;
>
> ## **[W1] Discuss Limitations of Criticism Mechanism**
> We evaluate the effectiveness of our method across a diverse range of tasks, covering both vision and language modalities (e.g., CIFAR10, CAR, Irony), multiple disciplines (e.g., Fashion, VQA-RAD), and human emotion understanding (e.g., Emotion, Irony). To further address the stability of the criticism mechanisms, we provide a systematic analysis in Section 4.2.1, where we derive three key insights into optimal configurations across broader task and model settings.
>
> We also discuss the influence of model scale in Section 4.2.1. Specifically, we treat the choice between black-box and white-box models as a practical consideration of model scale, since black-box models typically involve substantially larger parameter counts compared to their white-box counterparts. Given our motivation to leverage more capable large-scale LLMs to reduce human annotation workload, we intentionally select a substantially larger model as our annotator, rather than relying on smaller-scale models such as the 7B variant.
>
> &nbsp;
>
>
> ## **[W2] Expand Limited Experiment Scope**
> Here we address the two points mentioned in [W2] separately: (1) applicability to more complex/long-tail datasets, (2) multiple independent runs for stronger statistical validity.
>
> ### **(1) Applicability to More Complex/Long-tail Datasets**
> We agree that it is important to evaluate on more complex or long-tailed datasets. Therefore, we have added two additional experiments to address this concern:
>
> **[Allenoise]** Allenoise [https://github.com/allegro/AlleNoise] is a product classification dataset where similar category names often result in subjective and biased labelling. Due to time constraints, we focus on 20 labels under the "Sports & Travel" category, which exhibit approximately 70% label similarity—making the task ambiguous and prone to bias. Additionally, the largest category contains 27 times more samples than the smallest, reflecting a significant long-tail distribution. The results are shown below, demonstrating that ACT yields only a 1.4% performance gap in accuracy and a 0.0041 difference in F1-score compared to full human annotation.
>
> | **Data – Loss (AlleNoise – RoBERTa)**                     | **Accuracy (%)**   | **F1-score**         |
> |:----------------------------------------------------------|:------------------:|:--------------------:|
> | Human annotation – CE loss                                | 91.54 ± 1.84       | 0.8448 ± 0.0165       |
> | ACT data – ACT thre. loss (GPT4o self-criticism)          | 90.14 ± 1.79       | 0.8397 ± 0.0191       |
> | **Human–ACT performance gap**                             | 1.40               | 0.0041               |
> |
>
>
> **[CIFAR10-LT]** CIFAR10-LT [https://huggingface.co/datasets/tomas-gajarsky/cifar10-lt] is a long-tailed variant of CIFAR10 with an imbalance ratio of 100 between the most and least frequent classes. As shown below, ACT achieves a performance gap of 1.97% in accuracy and 0.0148 in F1-score compared to full human annotation.
>
> | **Data – Loss (CIFAR10-LT – ResNet)**                    | **Accuracy (%)**   | **F1-score**         |
> |:---------------------------------------------------------|:------------------:|:--------------------:|
> | Human annotation – CE loss                               | 70.29 ± 2.15       | 0.6820 ± 0.0261       |
> | ACT data – ACT thre. loss (GPT4o self-criticism)         | 68.32 ± 1.66       | 0.6672 ± 0.0263       |
> | **ACT – Human performance gap**                          | 1.97               | 0.0148               |
> |
>
> &nbsp;
>
> ### **(2)Multiple Independent Runs for Stronger Statistical Validity**
> We would like to clarify that our stability analysis indeed refers to multiple independent runs of the same experiment. As detailed in Appendix B (page 3 of the supplementary material), each experiment was conducted with 5 independent runs, and we report both the mean and standard deviation across these runs. Similarly, all results presented in Table 3 in the original paper are also averaged over 5 independent training runs, with standard deviations provided accordingly.
>
> &nbsp;
>
> ## **[W3] Explain Theoretical Assumption Issue**
> We thank the reviewer for pointing out the distinction between the theoretical assumption of perfect human labels and the realities of practical deployment. We agree that this issue can arise in real-world applications, especially for tasks involving ambiguous or subjective labels. In general, fully human-annotated data is considered the highest-quality supervision we can obtain, and it is standard to treat such labels as ground truth. However, this may not prevent the possibility of errors introduced by human annotators. Accordingly, we would like to emphasize that this is not a limitation specific to our method, but rather an inherent limitation of human annotation itself.
>
> &nbsp;
>
> ## **[W4 & Q5] Discuss Social & Ethical Considerations**
> We recognize a potential ethical concern that ACT, by reducing manual annotation workload, could impact traditional annotation jobs. However, ACT is designed to address the scalability challenges of human annotation, not to replace human roles. It enables human–AI collaboration by reserving human oversight for high-risk or ambiguous cases. Moreover, ACT may reduce the psychological burden of labelling harmful content (e.g., violence, nudity, etc.) by limiting human involvement to key instances. We will elaborate on these aspects in the revised paper.
>
> &nbsp;
>
> # **[Q1] Model Selection & Generalization**
>
> Thank you for the question, we will answer the two questions separately: (1) ACT's dependence on model types and scales, (2) test ACT on open-domain or real-world long-tail datasets.
>
> ### **(1) ACT's Dependence on Model Types and Scales**
> **[Specific Model Types]** Please refer to the user guideline provided in Figure 1. For most tasks, a general-purpose approach is to directly use GPT-4o self-criticism (vanilla version), which already reduces the downstream performance gap to within 2% while saving up to 90% of human annotation effort. This method does not rely on any specific model. For users seeking optimal performance, the Pro version allows for model selection to further improve results.
>
> **[Model Scales]** Please see response to [W1].
>
>
> ### **(2) Test ACT on Open-domain or Real-world Long-tail Datasets**
>
> Has the ACT pipeline been tested on open-domain datasets or real-world long-tail tasks for better generalization? The short answer is **YES**. We have provided empirical results on public long-tailed datasets under "[W2] Expand Limited Experiment Scope".
>
> &nbsp;
>
> ## **[Q2] Criticism Mechanism Improvements**
>
> **[Difference betweem Self- and Cross-Criticism in Complex Tasks]** Using AlleNoise as an example, we compare GPT-4o self-criticism with GPT-4o–Gemini1.5P cross-criticism. As shown in the table below, both approaches yield very similar downstream performance, with cross-criticism providing a slight improvement of 0.17% in accuracy and 0.001 in F1-score.
>
> | **Data – Loss (AlleNoise – RoBERTa)**                              | **Accuracy (%)**   | **F1-score**         |
> |:-------------------------------------------------------------------|:------------------:|:--------------------:|
> | ACT data – ACT thre. loss (GPT4o self-criticism)                   | 90.14 ± 1.79       | 0.8397 ± 0.0191       |
> | ACT data – ACT thre. loss (GPT4o–Gemini1.5P cross-criticism)       | 90.31 ± 1.57       | 0.8407 ± 0.0133       |
>
> &nbsp;
>
> **[Methods to Further Improve Criticism]** We see several promising directions for extension:
> - As discussed in Appendix G.2, we can build a multi-agent criticism system using strategies like majority voting, debate, or peer discussion among different models.
> - We can enhance the criticizer with ICL by providing examples of known annotation errors to guide its judgment.
> - For more complex tasks, RAG can be used to supply the model with relevant external knowledge for more informed criticism.
>
> &nbsp;
>
> ## **[Q3] Trade-offs in Annotation Quality**
> To clarify, as discussed in Section 4.1 (Insight 2), our findings do not suggest that Chain-of-Thought (CoT) reasoning is ineffective for annotation in general. Instead, we observed that its benefits may depend on the specific annotator model used. Specifically, for GPT-4o—the annotator we selected due to its consistently superior annotation performance—we observe that CoT reasoning does indeed provide measurable benefits. Thus, further optimization of CoT prompts may lead to additional improvements in annotation quality.
>
> &nbsp;
>
> ## **[Q4] Sampling Strategy Selection**
> We thank the reviewer for the comment and offer the following clarifications:
>
> **[Thresholding is Preferred]**: As noted in line 297, the threshold-based strategy is simple and involves only one parameter, which can be set according to the available human annotation budget. For example, with 10,000 samples and a 10% review budget, the threshold can be set at the 90th percentile of predicted error probabilities.
>
> **[Human Budget is Easy to Estimate]**: As discussed in Section 5.2, model accuracy—and thus human review needs—can be estimated from 100–300 labelled samples. For example, with 80% annotator accuracy, a 20% review budget suffices, assuming the criticizer catches all errors.

---

> > ### Comment · Reviewer_CvVf · 2025-08-04
> >
> > Thank you for your comprehensive rebuttal. Your responses have clarified several key concerns I had about your work.
> >
> > The additional experiments on long-tail datasets, clarification about multiple independent runs, expanded ethical considerations, and comparison between self-criticism and cross-criticism mechanisms help strengthen your paper. While there remain opportunities to further explore the stability of criticism mechanisms across more diverse tasks and model scales, your current results showing reasonable performance gaps compared to human annotation while significantly reducing annotation effort are promising. Based on these clarifications and the additional experimental results, I'm inclined to reconsider my initial assessment, as your approach appears to offer both theoretical insights and practical benefits for real-world annotation scenarios.

---

### Official Review · Reviewer_zzKU · 2025-07-02

**Clarity:** 2
**Significance:** 3
**Originality:** 2
**Rating:** 4
**Confidence:** 4

**Summary:**

This paper proposes the "Annotation with Critical Thinking" (ACT) pipeline to address the high cost of human data annotation. The method uses one Multimodal Large Language Model (MLLM) as an annotator and a second as a "criticizer" to identify potentially incorrect labels, strategically guiding limited human review to the most suspicious cases. Key contributions include validating the approach across diverse multimodal domains, offering practical implementation guidelines from empirical studies, and introducing a theoretically-grounded "ACT loss" function to ensure robust downstream training. Experiments demonstrate that this pipeline can achieve performance within 2% of models trained on fully human-annotated data, while reducing human annotation costs by up to 90%.

**Questions:**

1.What is the content of the Annotation Prompt? How does it guide the mllm to perform 1-0 label annotation on the input image?

2.Can the proposed method be extended to the training process of multimodal large models, not just for downstream tasks?

3.How would the ACT pipeline, particularly the criticizer's "error probability" estimation, be adapted for more subjective or generative tasks like rich image captioning or text summarization.

**Ethical Concerns:**

["NO or VERY MINOR ethics concerns only"]

**Final Justification:**

I think the authors did a good job addressing my concerns. After reading other reviews‘ comments and the responses, I decide to raise my score.

**Limitations:**

yes

**Quality:**

3

**Strengths And Weaknesses:**

Strengths：
1.The paper introduces specific metrics (AQG and ABS) to quantitatively assess the pipeline's efficiency at different budget levels. This provides a more nuanced performance analysis than a single accuracy score.

2.The proposed ACT incorporates the error probabilities of the critics, thereby providing a structured approach for training on mixed-quality data.

3.The validation across six recent MLLM families, covering both black-box and white-box types, provides a strong empirical basis for the paper's findings.


Weakness：
(1)The article is a bit hard to read, includes too many notations, and lacks details about the Figure 1 pipeline. This prevents the reader from having a clearer understanding of the proposed method.

(2)The evaluation is confined to tasks with simple categorical or short-text outputs. The framework's applicability to more complex, structured annotation tasks like object detection or semantic segmentation remains unexplored.

(3)While the paper focuses on saving human effort, it omits an analysis of the computational and API costs required to run two large MLLMs over entire datasets. A full cost-benefit analysis is necessary to assess the pipeline's true economic viability.

(4)The experiment was insufficient. The downstream validation is limited to a standard architecture (e.g., ResNet-18 for vision tasks ). The claims would be more compelling if the ACT-annotated data were tested on a broader range of models, including both modern Transformer-based architectures (e.g., ViT, CLIP) and models of a larger scale.

(5)The paper lacks ablation experiments that will examine how annotator/critic choice affects final downstream model performance.

(6)The experiments in Table 3 lack a comparison with existing baselines from related fields such as semi-supervised or traditional active learning, and further comparison would enhance the effectiveness of the method.

---

> ### Author Rebuttal · Authors · 2025-07-30
>
> Dear Reviewer zzKU,
>
> We sincerely thank you for the valuable suggestions regarding improving our experiments and extending our method. In response, we have (1) included 4 additional experiments/analyses, such as ablation studies and cost analysis to further demonstrate the effectiveness of our approach, and (2) discussed the potential extension of our method to broader tasks, including its role in MLLM training and subjective tasks. We will incorporate these improvements into the revised paper. Please see below for detailed additions and modifications.
>
> &nbsp;
>
> ## **[W1] Notations & Understandability**
>
> Thank you for your comment on notations and clarity. We first want to highlight that the notations are necessary for defining the key metrics (i.e., AQG, ABS) as well as facilitating our theoretical analysis in Section 5. In our revised version, we will (1) include a notation table for better reference, and (2) refine the caption of Figure 1, including an explanation of our ACT pipeline.
>
> &nbsp;
>
> ## **[W2] More Complex Structured Tasks**
>
> Thank you for your question. In our paper, we consider any task with a finite space of labels (as defined in Section 2), which includes many common tasks such as image classification, text classification, and VQA. These tasks are of critical importance because they require substantial annotation costs.
>
> However, we believe our proposed framework can be naturally extended to deal with more complex structured tasks that also have finite label sets, such as object detection and semantic segmentation.  For semantic segmentation, each pixel’s label still comes from a finite set, and our method can flag potential error regions, allowing human to correct those areas and therefore lower the cost significantly. For object detection, confidence scoring of candidate bounding boxes lets human focus on boxes with low confidence or prediction inconsistencies.
>
> We believe this strategy can be generalized to other tasks, and we will add more discussions in our revised version.
>
> &nbsp;
>
> ## **[W3] Cost-Benefit Analysis**
> We present a cost-benefit analysis using CIFAR10 (50k samples) to compare three annotation strategies: (1) 100% human annotation, (2) ACT with GPT4o self-criticism, and (3) ACT with GPT4o–Qwen2.5VL cross-criticism. As shown in the table below, API-based approaches are the most cost-efficient. Note that we exclude the cases where GPUs are not rented but owned, whose costs are difficult to estimate.
>
> We would also like to highlight that monetary costs are not the only concern. Human annotation also incurs significant time and educational costs, further highlighting the efficiency of our machine-based approach.
>
> | **Item** | **Human-only** | **ACT (GPT4o–GPT4o)** | **ACT (GPT4o–Qwen2.5VL)** |
> |----------|----------------|------------------------|----------------------------|
> | **GPT4o Annotation**  • API costs (OpenAI) | – | ~200 tokens × $10/1M = $0.002/image. Total = 50,000 × $0.002 = **$100** | Same as left → **$100** |
> | **GPT4o CoT Criticism**  • API costs (OpenAI) | – | ~200 tokens × $10/1M = $0.002/image. Total = 50,000 × $0.002 = **$100** | – |
> | **Qwen2.5VL CoT Criticism**  • GPU rent costs (RunPod) | – | – | 8 A100 GPUs × 144 hrs × $1.74/hr ≈ **$2000** |
> | **Human Annotation / Correction**  • Human costs (AWS avg.) | 50,000 images × $0.08 = **$4000** | 10% × 50,000 = 5,000 images → 5,000 × $0.08 = **$400** | Same as left → **$400** |
> | **Total Costs** | **$4000** | **$600** | **$2500** |
>
> &nbsp;
>
> ## **[W4] Broader Downstream Model Structures**
>
> We have tested the ACT-annotated data on a broader range of models, including an example of modern Transformer-based architectures (ViT) and another example of models with a larger scale (Qwen2.5 7B). The results can be found below. The small performance gap of 0.73% and 1.79% shows the generalizability of our method to a wide range of downstream structures.
>
> | **Data – Loss**                       | **Accuracy (CIFAR10-ViT)** | **Accuracy (Irony-Qwen2.5 7B)** |
> |--------------------------------------|-----------------------------|----------------------------------|
> | Human only – CE loss                 | 88.76 ± 0.08                | 77.17 ± 1.69                     |
> | Machine only – CE loss               | 86.59 ± 0.19                | 70.53 ± 1.53                     |
> | ACT data – ACT thre. loss            | 88.03 ± 0.16                | 75.38 ± 1.77                     |
> | **Human–Machine performance gap**    | 2.17                        | 6.63                             |
> | **Human–ACT performance gap**        | 0.73                        | 1.79                             |
> | **Human budget**                     | 11.52%                      | 17.98%                           |
>
>
>
> &nbsp;
>
> ## **[W5] Ablation of Annotator-Critisizer Selection**
> We have added ablation experiments that examine how annotator/criticizer choice affects final downstream model performance. The experiments are conducted with Cars, because variations in self- and cross-criticism performance are large on this dataset, based on Figure 2 in the paper. So, we would like to know whether the downstream performance also varies a lot.
>
> We find that the effectiveness of criticism is indeed related to downstream performance. The best-performing criticizer (highest ABS)  yields the smallest performance gap. However, the default GPT4o also performs reasonably well, keeping the gap within 2%.
>
> | **Cars – GPT4o Annotation**         | **GPT4o** | **Gemini1.5P** | **Claude3.5S** | **LLAVA-OV** | **Qwen2.5VL** | **InternVL2.5** |
> |:------------------------------------|:--------:|:--------------:|:--------------:|:------------:|:-------------:|:---------------:|
> | Criticism ability – ABS             |  60.6    |   **70.9**     |     53.2       |     55.9     |     54.4      |      57.5       |
> | ACT data – ACT thre. loss           | 86.00 ± 0.26 | **86.21 ± 0.31** | 85.21 ± 0.35 | 85.39 ± 0.29 | 85.80 ± 0.29 | 85.45 ± 0.31 |
> | Human–ACT performance gap           |   1.88   |    **1.67**    |      2.67      |     2.49     |     2.08      |      2.43       |
>
>
> &nbsp;
>
>
> ## **[W6] Comparison with More Baselines**
> Here we present the results of comparison with more existing baselines from related fields on ResNet18-CIFAR10:
>
> - **[Pseudo Labelling]** We randomly label 11.52% of the data to fine-tune ResNet18, then iteratively add high-confidence predictions (e.g., logit prob. > 0.9) as pseudo labels to the training set.
> - **[Active Pseudo Labelling]** We start by labelling 100 samples per class and progressively select uncertain samples (based on prediction entropy) for annotation until the human budget is exhausted.
>
> The results shown in the Table below indicate that our ACT approach outperforms these two methods by 4.28% and 2.56% in terms of performance gap from 100% human annotation.
>
>
> | **Method**                         | **Accuracy (%)**        | **Human Performance Gap (%)** |
> |:----------------------------------|:-------------------:|:-------------------------:|
> | Human-only – CE loss              | 88.66 ± 0.97        | –                         |
> | ACT – ACT thre. loss              | 87.95 ± 0.35        | 0.71                      |
> | Pseudo labeling – CE loss         | 83.67 ± 0.32        | 4.99                      |
> | Active Pseudo labeling – CE loss  | 85.39 ± 0.91        | 3.27                      |
>
> &nbsp;
>
> ## **[Q1] Annotation Prompt**
> We provide the detailed annotation prompt in Section C (page 4) of our Appendix. To guide the LLM in performing 0–1 label annotation, we directly prompt it using a structured format that includes the label list alongside the corresponding image data. This prompt design enables the model to associate relevant visual features with the appropriate binary labels effectively. For example, the naive annotation prompt of binary VQA is:
> #### Prompt:
> "{image_data} Answer the question based on the given image. The question is: {question}. The required output format is [0] for No and [1] for Yes. Do not return other text."
> #### Output
> "[answer_index]"
>
> &nbsp;
>
> ## **[Q2] Extension to MLLMs**
>
> As discussed in Weakness 4, we extend our approach beyond downstream tasks by applying it to the supervised fine-tuning (SFT) stage of LLMs, demonstrating its potential as a generalizable open-world annotator. Specifically, we annotate the Irony dataset using our method and fine-tune Qwen2.5-7B on this annotated data. The results show that training with ACT-annotated data leads to only a 1.89% drop in accuracy compared to training with human annotations, while achieving a 4.85% improvement over training with machine-generated annotations.
>
>
> &nbsp;
>
> ## **[Q3] Adaptation to Subjective or Generative Tasks**
>
>
> We appreciate the question on subjective or generative tasks. For tasks where the notion of “correct/incorrect” becomes subjective, the criticizer can output a continuous quality score or a relative ranking rather than a binary error probability.
> * The criticizer can leverage a strong evaluator, e.g., an MLLM‑based evaluator or a reward model, to detect semantic inconsistencies, factual mistakes, or language issues in each candidate output and return a numerical score or rank.
> * For content‑dense tasks such as multi‑fact text summarisation, we can first segment the summary into individual factual statements and then evaluate each statement separately. This allows human to focus only on the segments flagged as uncertain, rather than re‑checking the entire summary.
>
>
> We will incorporate a discussion of these extensions in the revised version.

---

### Official Review · Reviewer_sK1A · 2025-07-02

**Clarity:** 3
**Significance:** 3
**Originality:** 2
**Rating:** 5
**Confidence:** 3

**Summary:**

This paper presents a data annotation pipeline that combines Multimodal LLMs and human reviewers to reduce annotation cost while maintaining the quality of annotations. The core idea is to use an LLM to generate labels and another LLM to act as a criticizer, estimating the probability that a given label is incorrect. Human annotations are required only on the most suspicious cases, minimizing manual effort.

The authors show that it generalizes to NLP, vision, and multimodal tasks, and that with appropriate sampling strategies and a custom loss function derived from active M-estimation, downstream model performance can be brought within 2% of full human supervision while reducing human annotation load by up to 90%.

**Questions:**

- The paper sets the human review budget equal to the LLM’s error rate, but that assumes prior knowledge of how often the LLM is wrong. In a real-world scenario where no ground-truth labels exist, how would you recommend setting this budget? Could it be estimated dynamically?

**Ethical Concerns:**

["NO or VERY MINOR ethics concerns only"]

**Final Justification:**

Thanks to authors and reviewers for the discussion. I would keep my current score.

**Limitations:**

No. While the authors mention the limitations in the paper checklist, they do not have a limitation section. It is strongly recommended to create an additional limitations sections and discuss the point that authors already mentioned in the checklist.

**Paper Formatting Concerns:**

-

**Quality:**

3

**Strengths And Weaknesses:**

### Strengths

- The annotation setup is very well described and is easy to follow. It’s practical and general-purpose, as it can be applied to many annotation tasks.
- The authors test on image, text, and multimodal datasets, and show that the method works broadly.
- The experimentation is done quite thoughtfully, comparing different models, prompting strategies, and error estimation techniques. The results support the main claims well. The insights from each experiment is also clearly stated. Also, the separate strategies for black-box and white-box models were also very intuitive and well thought.
- The modified loss function smartly balances between human-verified and LLM-generated labels using estimated error probabilities, allowing the model to learn effectively even when most labels are noisy.


### Weaknesses
- The core idea, use LLMs to annotate and then review, is an incremental refinement of existing human-in-the-loop or LLM-as-judge pipelines. While the execution is polished, it is not conceptually novel.
-  The effectiveness of the loss adjustment depends heavily on how well the LLM criticizer estimates label errors and on the choice of sampling parameters, which may not generalize easily across tasks or models.
- As far as I understand, the paper assumes the ideal human review budget equals the LLM’s error rate, which means if GPT-4o is 88% accurate, then 12% of samples should be reviewed. But in real-world settings, we don’t know the true error rate without already having human labels which defeats the purpose. If the budget is set too low, important errors may be missed; if it’s too high, it wastes human effort. The framework doesn’t offer a way to estimate or adapt the budget dynamically, which limits its practicality.

---

> ### Author Rebuttal · Authors · 2025-07-30
>
> Dear Reviewer sK1A,
>
> We sincerely thank you for giving our paper a high score of 5. We’re glad to hear that you found our annotation setup clear and easy to follow. In particular, your comment on human budget estimation in practice was especially insightful and inspiring. In response, we clarified how annotator accuracy can be reliably estimated using only 100–300 labelled samples, which can also guide model selection. Additionally, we added a discussion on dynamic budget adjustment as a promising direction for future work.
>
>  &nbsp;
>
> ## **[W1] Concept Novelty**
> We would like to respectfully clarify a potential misunderstanding. While our ACT data pipeline indeed builds upon concepts related to human-in-the-loop and LLM-as-judge frameworks, our primary contribution does not lie in proposing a new technical paradigm. Instead, our core novelty lies in providing a **systematic exploration and theoretical foundations** on how to effectively integrate LLMs into the data annotation process to efficiently reduce human cost without compromising downstream training performance. For example:
>
> - **[Guidelines on Choosing Annotator & Criterizer]** We show concrete guidelines on how to choose suitable annotators and criticizers for different tasks and datasets.
> - **[Novel Evaluation Metrics]** We propose two new metrics: AQG and ABS to quantify annotation quality improvements and human budget efficiency. These metrics offer a novel way to assess annotation pipelines beyond traditional accuracy.
> - **[Theoretical Foundations]** We provide a detailed analysis of the ACT loss, showing that models trained on ACT-annotated data can achieve remarkable performance compared to those trained on fully human-annotated data. Our formulation is supported by theoretical guarantees on loss unbiasedness and parameter convergence.
>
> These contributions demonstrate that ACT is not merely a refinement of existing ideas, but rather a practical and theoretically grounded framework for scalable, high-quality data annotation. To better communicate this, we will revise the abstract to more clearly emphasise these innovations.
>
>
>
>  &nbsp;
>
>
> ## **[W2] Generalizability of Criticism & Sampling**
> We appreciate the reviewer’s thoughtful concern about the generalizability of criticism and sampling. Please allow us to offer the following explanation, based on our empirical observations and design considerations.
>
> - **[Generalizability of LLM-based Criticism]** While our theoretical analysis does indicate that the performance of the criticizer plays an important role in downstream effectiveness, we find that current LLMs already provide reasonably strong judgment capabilities in practice. Across a range of datasets, the criticizer consistently achieves over 50% recall in identifying machine annotation errors, many of which are inherently ambiguous or challenging. This results in a 5%–20% improvement in annotation accuracy over purely machine-generated labels. We are optimistic that as LLMs continue to advance, their criticism abilities will generalize even more effectively across diverse domains.
> - **[Easy Selection of Sampling Parameters]** Our recommended threshold-based sampling strategy only has one parameter, the threshold. It can be directly determined by the human annotation budget. We further explain how this budget can be easily estimated in practical applications in the response to weakness 3 and question 1 [W3 & Q1].
>
>
>
>  &nbsp;
>
>
>
> ## **[W3 & Q1] Human Budget Estimation in Practice**
> We appreciate the reviewer’s observation regarding the challenge of setting an appropriate human review budget in the absence of prior labels. Please allow us to clarify how this issue is addressed in our design and empirical workflow:
>
> - **[Annotator Accuracy Can be Estimated with Minimal Human Efforts]** We agree with the reviewer that the exact error rate of an LLM annotator cannot be determined without some human labels. However, this does not contradict our goal of reducing human effort. In practice, we find that annotator accuracy can be reliably estimated using a small number of labelled samples—typically between 100 and 300, which represents only a tiny fraction of the dataset. This lightweight supervision is sufficient to guide the estimation of an appropriate human budget. In addition, as we have shown in Appendix F.2, slightly increase this budget by about 10% can effectively safeguard downstream performance across most datasets.
> - **[Labelled Data for Budget Estimation Can Also be Used for Model Selection]** Furthermore, as discussed in Section 4.3, although GPT-4o already performs reasonably well under self-criticism, choosing the most suitable annotator–criticizer pair for a given task typically requires a small set of human-labelled samples. The labelled data are thus also used for the model selection process. This dual-purpose design avoids additional human annotation efforts while improving the overall performance of the ACT pipeline.
> - **[Dynamic Budget Adjustment is a Promising Direction for Future Improvement]** Finally, we agree that dynamically estimating and adjusting the human budget during the annotation process is a meaningful and practical strategy. As more human-verified labels are accumulated over time, it becomes increasingly feasible to refine the error rate estimation and update the budget allocation accordingly. We will incorporate this discussion into the paper to highlight the potential for further improving the adaptability of the ACT pipeline.
>
>  &nbsp;
>
>
> ## **[Limitation]**
> Thank you very much for pointing this out. We will add a limitation section in our revised paper.

---

> > ### Comment · Reviewer_sK1A · 2025-08-06
> >
> > Thanks to authors for addressing my comments. I believe my scores still reflect my overall assessment about the paper.

---

### Official Review · Reviewer_MY7W · 2025-07-03

**Clarity:** 2
**Significance:** 2
**Originality:** 2
**Rating:** 5
**Confidence:** 4

**Summary:**

This paper proposes ACT (Annotation with Critical Thinking), a general-purpose, LLM-powered data annotation framework that aims to reduce human annotation costs while maintaining high label quality across diverse domains. In ACT, a multimodal large language model (VLM) first serves as the annotator, generating initial labels. A second VLM acts as a critic, estimating the likelihood that each label is incorrect. Only those samples with high estimated error probability are then reviewed by human annotators, allowing strategic use of limited human resources.

The paper evaluates ACT across six datasets spanning image classification, text classification, and visual question answering, and demonstrates that ACT can reduce the human annotation effort by up to 90%, while achieving downstream model performance within 2% of fully human-annotated baselines.

**Questions:**

1. The paper mentions that self-criticism performs competitively with cross-criticism. It would be helpful if the authors could provide statistical evidence showing how often the critic, whether self or cross, correctly identified annotation errors. Reporting the frequency of true error detections by the critic across different settings would strengthen this claim.

2. The paper would benefit from a deeper analysis of cases where the annotator provides a label, the critic flags it as incorrect, but a human evaluator determines that the annotator was actually correct. Including such examples and their frequency can offer valuable insight into the critic's reliability and limitations in ambiguous or edge cases.

3. While the experiments are conducted on relatively simple datasets like Fashion-MNIST and CIFAR, it remains unclear how the approach would perform in more complex tasks where label ambiguity and bias are more prominent. Specifically, it would be useful to discuss scenarios where the annotator and critic share similar biases, or when their biases differ significantly, and how that affects overall performance. Can they identify tasks or setup, where maybe more human intervention is required and to what degree human budget will change.

4. The current setup treats the annotator as fixed. Would the annotation quality improve if the annotator received feedback from the critic after each generation? It would be interesting to evaluate this in both self-critique and cross-critique conditions. This feedback loop could potentially enhance annotation quality over time and is worth exploring.

**Ethical Concerns:**

["NO or VERY MINOR ethics concerns only"]

**Final Justification:**

I am satisfied with the experiments provided. Especially that the approach scales to noisy and biased dataset, making the overall contribution much stronger.

**Limitations:**

yes

**Quality:**

2

**Strengths And Weaknesses:**

Strengths:

Practical and Scalable Annotation Pipeline: The ACT framework offers a practical solution to reduce human annotation effort by combining LLM-generated labels with a critic model that flags potentially erroneous annotations. This pipeline is model-agnostic, training-free, and scalable across domains including vision, language, and multimodal tasks.

Strong Empirical Results with High Cost Efficiency: The framework achieves near-parity with fully human-labeled data (within 2% accuracy gap) while saving up to 90% of human effort. The results are consistent across diverse benchmarks such as CIFAR-10, Fashion-MNIST, VQA, and text classification tasks, validating both the quality and efficiency of the method.

Insightful Analysis: The paper provides detailed empirical insights (e.g., when CoT helps, self vs. cross-criticism).

Weaknesses:

Limited Evaluation on Complex or Biased Tasks: The experiments are limited to relatively simple datasets like Fashion-MNIST and CIFAR. It is unclear how well the framework generalizes to more complex tasks, particularly those with noisy or biased labels. The paper does not address how annotator–critic interactions behave when both agents share similar biases or when their biases are orthogonal, which could significantly impact performance in real-world settings.

Insufficient Insight into Annotator–Critic Disagreements: There is limited discussion of cases where the critic flags an error but a human adjudicator ultimately agrees with the annotator. These disagreement scenarios are crucial for understanding the critic’s reliability and could reveal whether critics tend to overcorrect or introduce new types of errors.

Potential Bias Propagation to Downstream Models: Since the ACT pipeline relies heavily on VLM-generated annotations and corrections, there is a risk that any systematic biases in these large models will propagate to the final annotated dataset and subsequently to smaller downstream models trained on it. The paper does not address how such biases are detected, mitigated, or evaluated.

---

> ### Author Rebuttal · Authors · 2025-07-30
>
> Dear Reviewer MY7W,
>
> We sincerely appreciate the time and effort you devoted to evaluating our work. We are especially inspired by your 4th question regarding the annotation feedback loop, which provides a valuable direction for our future work. In the following, we address your concerns by: (1) adding a new case study to evaluate ACT’s robustness to bias and noise, (2) providing deeper analyses of critic error capture and overcorrection, and (3) discussing potential bias propagation and directions for improvements.
>
> &nbsp;
>
> ## **[W1 & Q3] Complex Task & Critic Bias**
> Evaluating robustness of ACT under noisy or biased scenarios is indeed a good direction. We address this concern in two parts: (1) a case study on noisy & biased dataset AlleNoise [https://github.com/allegro/AlleNoise] (2) an explanation of how bias was reflected in our original experiments.
>
> &nbsp;
>
> ### **(1) Case Study on Noisy & Biased Data**
> **[Data & Task Description]** AlleNoise is a product classification dataset where similar category names often lead to subjective and biased labelling. Merchant-provided labels contain ~15% noise, while expert-annotated labels serve as unbiased ground truth. Due to time constraints, we test on 20 labels under "Sports & Travel" with ~70% label similarity, making it ambiguous and prone to bias. We try to explore if ACT downstream performance is close to the unbiased labels, which means ACT is robust to label ambiguity. We also compared the difference between self- and cross-criticism.
>
> **[Experiment Results: Can ACT be generalized to biased & noisy data?]** The short answer is **YES**. In the Table below, we can see that ACT data performance is very close to unbiased labels, and outperforms the biased labels by 11.94% in accuracy. Self- and cross-criticism show similar results.
>
> | **Data - Loss (AlleNoise - RoBERTa)**                        | **Accuracy (%)** | **F1-score**    |
> | ------------------------------------------------------------ | ---------------- | --------------- |
> | Biased merchant annotation - CE loss                         | 78.37 ± 1.18     | 0.6828 ± 0.0232 |
> | Unbiased expert annotation - CE loss                         | 91.54 ± 1.84     | 0.8448 ± 0.0165 |
> | GPT4o machine annotation - CE loss                           | 81.40 ± 2.09     | 0.7385 ± 0.0094 |
> | ACT data - ACT thre. loss (GPT4o self-criticism)             | 90.14 ± 1.79     | 0.8397 ± 0.0191 |
> | ACT data - ACT thre. loss (GPT4o-Gemini1.5P cross-criticism) | 90.31 ± 1.57     | 0.8407 ± 0.0133 |
>
> &nbsp;
>
> ### **(2) Bias Reflected in Our Original Experiments**
>
> **[Insights on Agent Bias]** In our original setup, self-criticism can be seen as a same-bias setting, while cross-criticism reflects different biases (from different model families). Based on this, we find that bias difference does not significantly impact performance, as both shown in the paper and the case study. This somehow alleviate the burden of model selection. Nonetheless, we also include bias-reduction strategies later in response to [W3] as a precaution, in case bias does impact downstream performance in very complex real-world scenarios.
>
> **[Impact of Bias on Human Budget]** Since human budgets can be affected by both subjective differences (bias) and predictive uncertainty (variance), we sugget focusing more on final annotation quality, not specific bias types. To estimate human effort, we recommend labelling a small subset (100-300) manually, then evaluating annotator error rate. For example, in VQA-RAD, due to the complexity of medical data, GPT annotator reached only ~70% accuracy, so we used a 30% human budget, which closed the downstream gap with 100% human (see Table 3).
>
> &nbsp;
>
> ## **[W2 & Q1 & Q2] Deeper Analyses on Critic True Positives & False Positives**
> Here we consider two situations and apply deeper analyses to the criticism results: (1) From W2 and Q2, the situation where critic flags an error but a human adjudicator ultimately agrees with the annotator (False Positives). (2) From Q1, the situation where the criticizer correctly identified annotation errors (True Positives). These results will be included in the revised paper to further enrich insights on the ACT data pipeline.
>
> &nbsp;
>
> ### **(1) Analyze Criticism Reliability with False Positives**
> The table below reports the number of false positives produced by different error detection methods across six datasets. The human budget is consistent with Table 3 in the main paper. We compare GPT4o self-criticism and the best-performing cross-criticizer for each task, with random error sampling included as a baseline. We observe the following:
>
> - MLLM-based error detection significantly outperforms random sampling, consistently yielding fewer false positives across tasks.
> - False positives account for approximately 5%–15% of the total dataset, depending on the complexity and ambiguity of the task.
>
> These results suggest that MLLM-based criticism is an effective strategy for identifying annotation errors. In addition, there remains room for improvement, especially through incorporating sample-specific analysis and adaptive selection mechanisms in real-world applications.
>
> | #False Positives |  |  |  |  |  |  |
> |-------------------|--|--|--|--|--|--|
> | **Error Detection Methods**             | **CIFAR10 (data size~50k)** | **Fashion (data size~60k)** | **Cars (data size~8k)** | **Emotion (data size~3k)** | **Irony (data size~3k)** | **VQA-RAD (data size~1k)** |
> | Random Sampling            | 4866                          | 9852                           | 704                        | 480                           | 640                         | 198                           |
> | GPT4o Self-Criticizer      | 2805                          | 6395                           | 594                        | 302                           | 591                         | 123                           |
> | Optimal Cross-Criticizer   | 2692                          | 6087                           | 514                        | 299                           | 455                         | 113                           |
>
> &nbsp;
>
> ### **(2) Compare Self- and Cross-Criticism with True Positives**
>
> We use dataset VQA-RAD as an example, where GPT4o only achieves 69.85% annotation accuracy. The human budget is set to 30.15%. In the following table, we present the number of True Positives and annotation accuracy after human correction. Randomly sampled errors are included as a baseline. It is clear that all MLLM criticizers perform much better than random sampling, while self- and cross-criticism do not show huge differences. This further strengthens the claim that self-criticism performs competitively with cross-criticism.
>
> | # True Positives and Final Annotation Accuracy |  |  |
> |-----------------------------------------------|--|--|
> | **Error Detection Methods**           | **#True Positives** | **Final Annotation Accuracy** |
> | Random Sampling           | 85                  | 78.89%                        |
> | GPT4o (Self)              | 160                 | 86.87%                        |
> | Gemini1.5P (Cross)        | 153                 | 86.12%                        |
> | Claude3.5S (Cross)        | 159                 | 86.76%                        |
> | LLaVA-OV (Cross)          | 158                 | 86.66%                        |
> | Qwen2.5VL (Cross)         | 170                 | 87.94%                        |
> | InternVL2.5 (Cross)       | 149                 | 85.70%                        |
>
> &nbsp;
>
> ## **[W3] Potential Bias Propagation to Downstream Mode**
>
> Indeed, our method can be viewed as a type of label distillation from large models. Thus, it is possible that systematic bias of VLM annotator and critic propagate to downstream tasks. In practice, such bias can be identified and mitigated via targeted cast analyses and prompt refinements accordingly. Besides, one can also leverage RAG to integrate a knowledge base that helps reduce bias. We will include more discussions in the revised paper.
>
> &nbsp;
>
> ## **[Q4] Annotation Feedback Loop**
>
> Thank you for this constructive suggestion. We believe this will be a valuable direction for our future work, which involves building an agentic system that inplements an annotation feedback loop. Such a system is expected to improve firt-run label quality over time and reducing reliance on human correction. We will include a discussion of this idea in Appendix G.

---

> > ### Comment · Reviewer_MY7W · 2025-08-04
> >
> > Thank you to the authors for their response. I am very pleased with the clarifications, especially seeing that the setup performs well on more challenging datasets. I also appreciate that my feedback created an opportunity to further explore the approach through a feedback loop. Based on the responses, I am inclined to increase my score.

---

### Note · Authors · 2025-08-14

Dear Area Chair and Reviewers,

We would like to sincerely thank you for your efforts and suggestions during the review process. We greatly appreciate the opportunity to engage with the reviewers, which has significantly contributed to the improvement of our paper. Most of the concerns raised have been carefully addressed in our responses.

All clarifications and additional results provided in our responses will be further revised and incorporated into the paper. We are always open to further discussions and welcome any additional comments or suggestions that could help us make our work even better.

Thank you again for your time and considerations.

Best regards,

Authors

---

### Decision · Program_Chairs · 2025-09-17

**Decision:**

Accept (poster)

**Comment:**

This paper proposes a data annotation pipeline built on an Annotation-with-Critical-Thinking (ACT) design that uses multimodal large language models as both annotators and critics to identify potentially incorrect labels, directing human review effort to fewer samples. The work provides some insights into how to modify the loss function, and demonstrates up to 90% reduction in human annotation costs while maintaining performance within 2% of fully human-annotated baselines across a few tasks (mainly in classification).

While the overall rating of this paper is ok, there are some points to be addressed to further strengthen the contribution:

(1) First, the experimental evaluation needs substantial expansion beyond simple datasets like Fashion-MNIST and CIFAR to include more complex and (potentially) biased tasks, particularly for subjective tasks like image captioning and open questions (from Reviewer MY7W, Reviewer CvVf, Reviewer zzKU).

(2) The framework requires better handling of bias propagation from the annotator and critic to downstream tasks, including methods to detect and mitigate systematic biases that could affect the final annotated datasets (from Reviewer MY7W, Reviewer sK1A).

(3) The analysis should include some in-depth discussion of annotator-critic disagreement cases and scenarios where both agents share similar biases, which is crucial for understanding critic reliability (from Reviewer MY7W, Reviewer sK1A, Reviewer CvVf).